# The gut microbiome and metabolome associate with *Schistosoma mansoni* infection and cardiovascular disease risk in Uganda

Bridgious Walusimbi [1,2,3,4] ✉, Melissa AE Lawson[4], Allison J. Bancroft[4], Jacent Nassuuna[1], Drupad K. Trivedi [5], George Taylor[6], Richard E. Sanya[7], Emily L. Webb [8], David P. Kateete[2,10], Richard K. Grencis [4,10] & Alison M. Elliott [1,9,10]

Helminth infections are consistently associated with reduced cardiovascular disease (CVD) risk, yet the biological mechanisms underlying this relationship remain unclear. The gut microbiome and metabolome are key regulators of cardiometabolic health and may mediate infection-associated effects on host physiology. Here we show that *Schistosoma mansoni* infection associates with distinct gut microbial and metabolic profiles linked to CVD risk in people living in Uganda. In a cross-sectional study of 209 individuals living in communities with contrasting *S. mansoni* endemicity, we profile the gut microbiome using 16S rRNA gene sequencing and the faecal metabolome using liquid chromatography–mass spectrometry. *S. mansoni* infection associates with increased gut microbial diversity and distinct taxonomic signatures, including enrichment of taxa such as *Treponema* and depletion of *Prevotella* and *Streptococcus*. Several infection-associated microbial taxa statistically mediate the relationships between *S. mansoni* infection and cardiovascular disease risk. Faecal metabolomic profiling identifies infection-associated metabolites, and integrative analyses showed linked microbe–metabolite networks associated with cardiovascular risk. These findings identify gut microbiome and metabolome signatures associated with *S. mansoni* infection and cardiovascular disease risk in Uganda. Although causality cannot be inferred, this work provides insight into host–parasite–microbiome interactions and highlights microbial and metabolic pathways relevant to cardiometabolic health.

Globally, cardiovascular diseases (CVDs) pose a significant threat to public health, consistently ranking as the primary cause of mortality over the last thirty years[1]. For example, CVDs accounted for approximately 20.5 million deaths (over 31% of global deaths) in 2021 alone[2]. More than 75% of CVD related deaths have been reported to occur in low and middle income countries, demonstrating a critical need for intensified research to address the risk factors for CVD that can be modulated to reduce the incidence of disease[3]. These risk factors include dietary risks, high systolic blood pressure, dyslipidaemia (particularly high low-density lipoprotein (LDL) cholesterol) and high fasting plasma glucose[1].

Several studies have linked cardiovascular risk factors with the immune system response, both in humans and animal models[4]. Particularly, chronic inflammation has been highlighted as the main immunological feature characterising cardiovascular or metabolic risk[5,6]. For example, increased circulating tumour necrosis factor (TNF)-α is associated with glucose intolerance, and inhibiting its expression in adipose tissues affects sensitivity to insulin, and tolerance for glucose, in obese individuals[6].

Additionally, inflammatory pathways involving the activation of macrophages, dendritic cells, and mast cells have been shown to rely on the availability of dietary lipids such as saturated fats and cholesterol[7]. Dietary lipids such as omega-3 fatty acids have been linked to production of inflammatory cytokines such as TNF-α and interleukin (IL)−2[8]. As such, this lipid-inflammation interplay suggests that inflammation alters lipid profiles and metabolism in hosts. Multiple lines of evidence show that cytokines such as IL-6, IL-1 and TNF-α are associated with increased production of triglycerides and LDL cholesterol levels in serum[6,9–11]. Atherosclerosis, a chronic inflammatory condition typified by thickening of arterial walls, mediated by accumulation of lipids and cells such as macrophages in the vascular intima, is central in cardiovascular disease prognosis[12]. Given the centrality of inflammation in cardiovascular risk, it has been hypothesised that infections that induce immunomodulatory responses protect hosts against metabolic disorders.

One such class of infections with immunomodulatory effects are chronic helminth infections[13]. These are characterised by a polarised T-helper 2 cell response, important in resisting or eliminating helminth infections in the host[14]. However, helminths can modulate the host's immune response to prolong their own survival[14–17]. For example, helminths induce production of IL-10, a cytokine known to be pivotal in suppressing inflammation, to enhance their survival[16,18–20]. Therefore, owing to their ability to downregulate inflammation in the host, the hypothesis that helminths may be protective against CVD risk is plausible.

Several epidemiological studies have shown an inverse association between chronic helminth infection and metabolic risk factors such as LDL cholesterol and high blood pressure. For example, Wiria et al., reported reduced total cholesterol, LDL cholesterol, body mass index (BMI) and waist-to-hip ratio in people infected with soil transmitted helminth (STH) compared to those without STH infection among Indonesian subjects living in helminth endemic region[21]. In another study by Magen et al, individuals with chronic Opisthorchis felineus infection had significantly reduced total cholesterol compared to those without the infection[22]. Furthermore, Shen et al found an inverse association between previous schistosomiasis infection and triglyceride levels, waist-to-hip ratio and BMI[23]. In the same study, diastolic blood pressure was significantly lower in subjects with previous schistosomiasis than those without infection[23]. Moreover, in separate cross-sectional studies investigating whether previous schistosome infection protects against development of diabetes and metabolic syndrome, Chen et al found lowered systolic blood pressure (SBP) and diastolic blood pressure (DBP) in people with schistosomiasis infection compared to those without[24].

The aforementioned evidence is further supported by meta-analyses that have begun to show the importance of the inverse association of helminths with CVD risk. Tracey et al reported an association of lower glucose levels, insulin resistance, metabolic syndrome, and a 50% reduced likelihood of susceptibility to CVD risk factors such as type 2 diabetes (T2D), with helminth infection[25]. This was reaffirmed by Rennie et al who found reduced fasting glucose, glycated haemoglobin (HbA1c) levels, prevalence of T2D and metabolic syndrome in people with helminth infections compared to those without[26].

Given that much of the existing evidence linking helminth infections to cardiometabolic protection is informed by studies focusing on STH, it would be informative to investigate how helminths acquired through alternative routes, such as S. mansoni, might affect one's cardiovascular risk, in a population with a high prevalence of S. mansoni infection, and in case of a protective effect, study the mechanisms by which these parasites bring about this benefit to the host[27]. Despite the well-reviewed importance of the anti-inflammatory effect of helminths such as S. mansoni in protecting the host against CVD, other possible pathways have been suggested[28].

Schistosomiasis, caused by parasitic trematodes of the genus Schistosoma, remains one of the most prevalent neglected tropical diseases worldwide[29], affecting millions of individuals, primarily in low-resource regions. Beyond its direct pathological effects, emerging evidence suggests a complex interplay between schistosomiasis infection and the modulation of host immune responses, metabolic pathways, and disease susceptibility, and more recently the potential of infection to have immune-mediated protective effect against cardiovascular disease risk[30–32]. However, the mechanisms underlying these potential benefits remain poorly understood.

One promising area of investigation is the gut microbiota, a complex ecosystem of trillions of microorganisms that profoundly shape host immunity, metabolism, and overall health[33,34]. Dysbiosis, characterised by abnormal changes in the composition and function of gut microbiota, has been implicated in various pathological conditions, including CVDs[34–36]. Helminth infections are known to modulate the gut microbiome, but it remains unclear whether such changes play a mediating role in the relationship between S. mansoni infection and cardiovascular risk.

There is growing evidence suggesting that the gut microbiota and the metabolites they produce serve as critical mediators of host-parasite interactions. In the context of schistosomiasis, the parasite-host interaction within the gut environment can influence microbial composition and metabolic activity, leading to systemic effects on host physiology and immune responses[37]. Moreover, specific microbial metabolites, such as short-chain fatty acids (SCFAs), and trimethylamine N-oxide (TMAO), have been implicated in modulating CVD risk factors such as blood pressure and cholesterol[33], and may contribute to the observed protective effect of schistosomiasis against CVD risk.

Despite the emerging evidence separately implicating the gut microbiome and its metabolites, and S. mansoni, the precise mechanisms underlying this complex helminth-microbiome interplay in driving cardiovascular risk modulation remain poorly understood. Our previous findings from a cluster-randomised trial involving 1,898 participants (the Lake Victoria Island Intervention Trial on Worms and Allergy-related Diseases [LaVIISWA], which was extended to investigate metabolic outcomes) showed that Schistosoma mansoni infection was associated with lower levels of total and LDL cholesterol, while intensive anthelmintic treatment led to an increase in LDL cholesterol[27]. Further, heavy and moderate S. mansoni infection intensities were associated with lower diastolic blood pressure, triglycerides and LDL cholesterol[27].

In pursuit of a deeper understanding of how S. mansoni infection could lead to these changes in CVD risk, the current work therefore used samples from our LaVIISWA trial and. Urban survey study, aiming at deciphering S. mansoni-associated alterations in gut microbial composition and diversity, the potential contribution of gut microbiome the observed associations with CVD risk factors, and the gut-microbiome and metabolome interaction in the context of chronic schistosomiasis infection and its impact on cardiovascular risk.

By using an integrative, multi-omics approach including microbiome and metabolomics, we unravel the molecular pathways and microbial signatures associated with the protective effect of schistosomiasis on CVD risk. Ultimately, this research provides novel insights into host-parasite interactions, microbial dysbiosis, and metabolic influence, with implications for the development of targeted interventions to mitigate cardiovascular disease risk in humans.

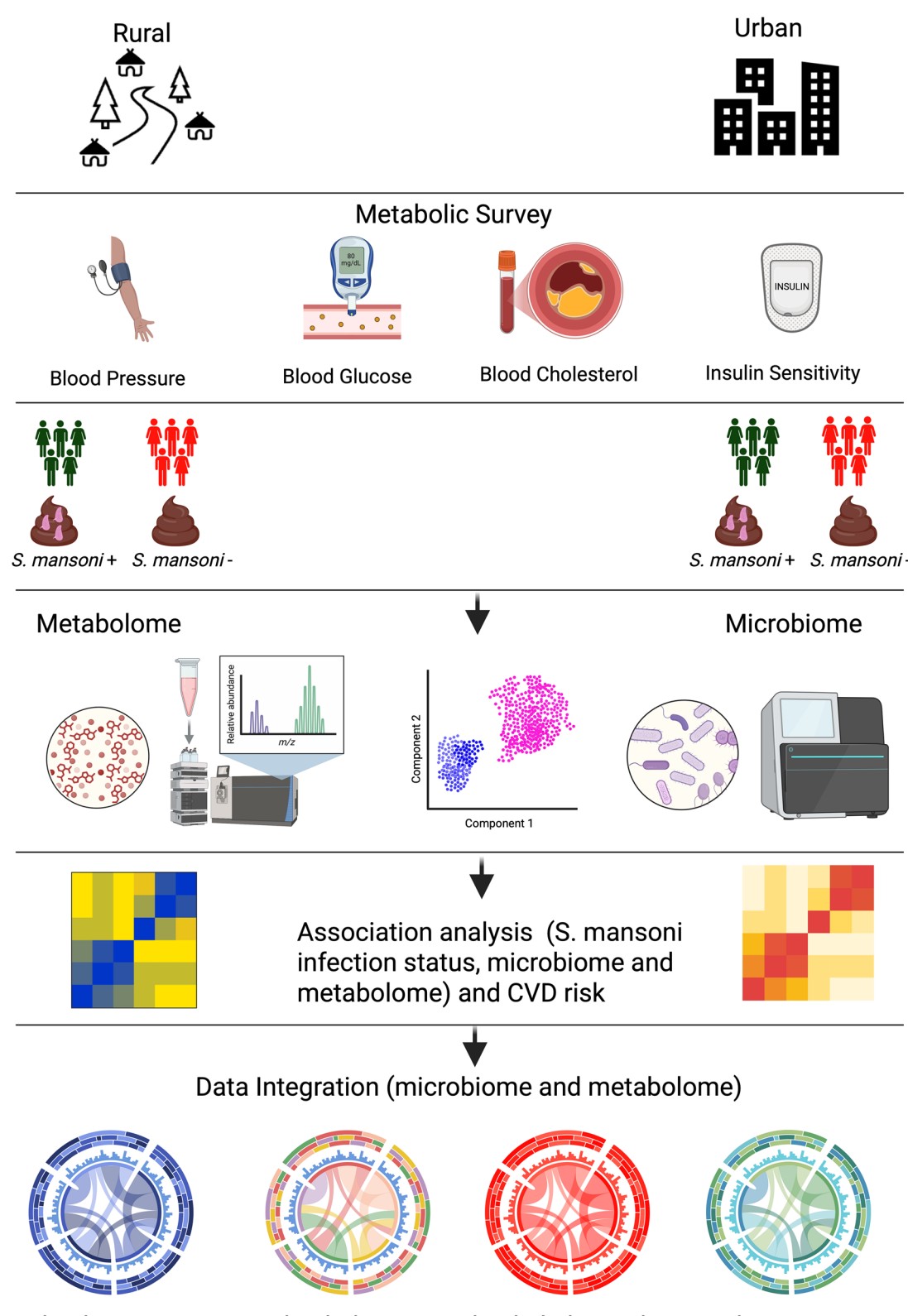

**Fig. 1 | Experimental and analytical flowchart.** Created in BioRender. Walusimbi, B. (2026) https://BioRender.com/a407o09.

## Results

Figure 1 provides an overview of the study design, participant selection, and analytical workflow, and downstream multi-omics and cardiovascular risk analyses.

As summarised in Table 1, this study included 209 participants selected based on the availability of lipid profiles (LDL, HDL, total cholesterol, triglycerides) and blood pressure data. Of these, 128 participants (61.2%) were classified as *Schistosoma mansoni* positive,

defined by positive results on both Kato-Katz microscopy and PCR, while 81 participants (38.8%) were negative on both tests.

The cohort comprised 108 males and 101 females. Among females, 43 were *S. mansoni* positive and 60 were negative. Participants were distributed across age groups: 10–19 years, 20–29 years, 30–39 years, and 40 years and above.

From the rural survey (LaVIISWA trial), 84 *S. mansoni* infected participants were included, with 43 of these from the intensive treatment arm. Among the negative individuals, 43 were from the rural setting and 38 from the urban setting. This distribution provides a balanced comparison across infection status, sex, age, and environmental exposure. A diet distribution analysis of the participants in rural communities showed that the majority (63%) were predominantly fish eaters, while 27.7% consumed a mixed diet. The other diet types were vegetarians and meat eaters that accounted for 5.9% and 3.4%, respectively, of the rural participants. The algorithm used for the diet distribution analysis is shown in Supplementary Fig. 1.

## S. mansoni infection is associated with altered gut microbial diversity and composition

We first performed *16S rRNA* amplicon sequencing on faecal samples from all the individuals used in this study. We found that sample gut microbial diversity (alpha diversity) was significantly higher in *S. m+* compared to *S. m-* ($p = 0.048$ and $p = 0.008$, for Shannon index and observed richness, respectively; Fig. 2A). On the other hand, Bray-Curtis-based beta diversity analysis did not show significant separation (PERMANOVA $p = 0.175$; Fig. 2B) between *S. m+* and *S. m-* individuals living in the rural setting, although a difference in the overall microbial community structure was observed between the *S. m+* and *S. m-* individuals living in the urban setting (PERMANOVA $p = 0.011$; Fig. 2C). In addition, we compared the beta diversity of participants that were under intensive anthelminthic treatment to those under standard treatment and we observed no difference in clustering (see Supplementary Fig. 2), suggesting that anthelminthic treatment was unlikely to meaningfully affect the microbiome-metabolome analyses or introduce bias when comparing rural and urban participants.

We next asked if there were specific bacteria genera that could be used to discriminate between *S. m+* and *S. m−* individuals. Linear

### Table 1 | Characteristics of the study participants from rural and urban communities

| Characteristics | Infected n = 128 | Uninfected n = 81 | Total n = 209 |
|---|---|---|---|
| Females/ Males | 42/86 | 59/22 | 101/108 |
| Age group (years) | | | |
| • 10-19 | 37 | 23 | 60 |
| • 20-29 | 30 | 25 | 55 |
| • 30-39 | 29 | 17 | 46 |
| • 40+ years | 32 | 16 | 48 |
| Setting | | | |
| • Rural | 84 | 43 | 127 |
| I. Intensive treatment | 43 | 32 | 75 |
| II. Standard treatment | 41 | 11 | 52 |
| • Urban | 44 | 38 | 82 |

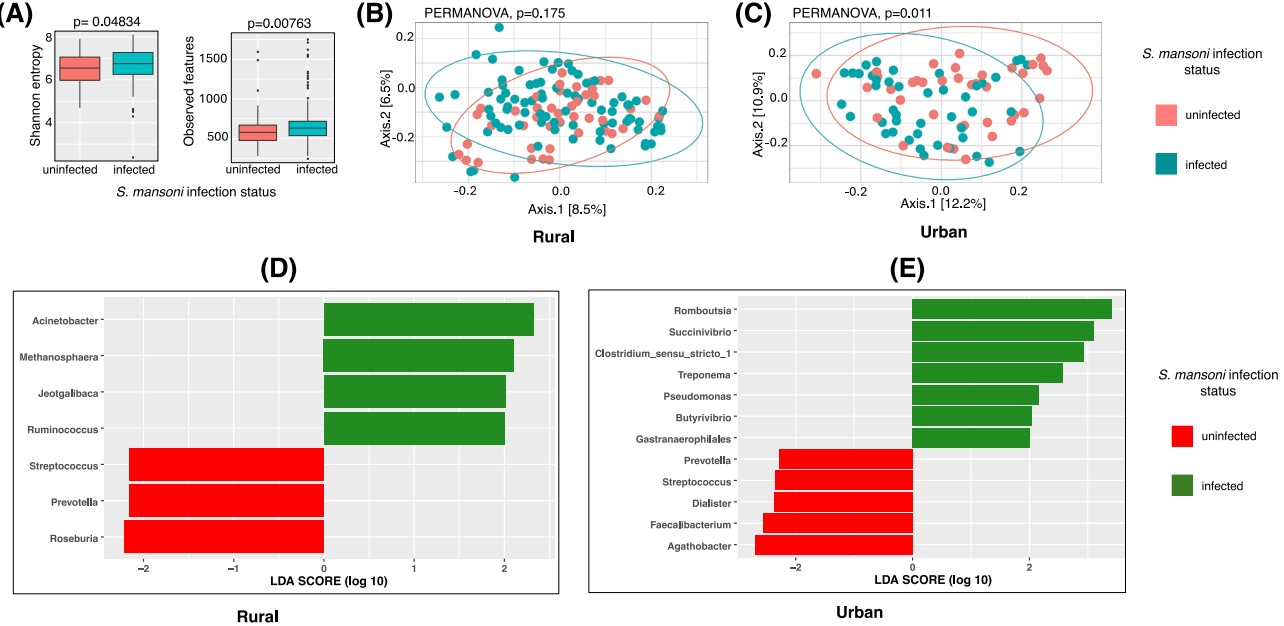

**Fig. 2 | Microbial diversity comparison of *S. mansoni* infected (*S. m +* ) and uninfected individuals (*S. m-*). A** Box plots comparing gut microbial alpha diversity between *S. mansoni*-infected (*S. m +*, *n* = 128) and uninfected (*S. m−*, *n* = 81) individuals (total *n* = 209), assessed using observed richness and the Shannon diversity index. For observed richness, *S. m+* individuals had a higher median richness (median = 632.5; interquartile range [IQR]: 537.0–718.0) compared with *S. m−* individuals (median = 579; IQR: 477–670). For the Shannon index, diversity was also higher in *S. m+* individuals (median = 6.751; IQR: 6.36–7.27) than in *S. m−* individuals (median = 6.548; IQR: 5.98–7.06). Box plots display the median (centre line), the interquartile range (bounds of the box; 25th–75th percentiles), and whiskers extending to the values within 1.5X IQR from the 25th–75th percentiles and points outside the range are plotted as outliers. Group differences were assessed using the Kruskal–Wallis test (observed richness: χ² (df = 1) = 7.12,

$P = 0.00763$; Shannon index: χ² (df = 1) = 3.90, $P = 0.048$). **B** Comparison of Beta diversity (using Bray-Curtis distance) in *S. m+* and *S. m-* individuals living in rural Uganda. These were similar (PERMANOVA p = 0.175, R² = 0.00923, F = 1.1646, Number of permutations= 999). **C** *S. m+* showing an overall microbiome structure that is different from the *S. m-* individuals (PERMANOVA, p = 0.011, R² = 0.02207 F =1.8055, Number of permutations= 999). **D** and **E** Linear Discriminant Analysis (LDA) scores of bacteria that are differentially abundant between (*S. m +* ) and (*S. m-*). LDA scores show the measure of effect of each genus. Bacteria with LDA score > 2 were differentially enriched in a particular group. **D** shows LDA results for *S. m+* and *S. m-* living in the rural setting while **E** shows LDA results for *S. m+* and *S. m-* living in the urban setting. The red bars represent microbes that are more abundant in the *S. m-* individuals while the green bars show those that are more abundant in the *S. m+* individuals.

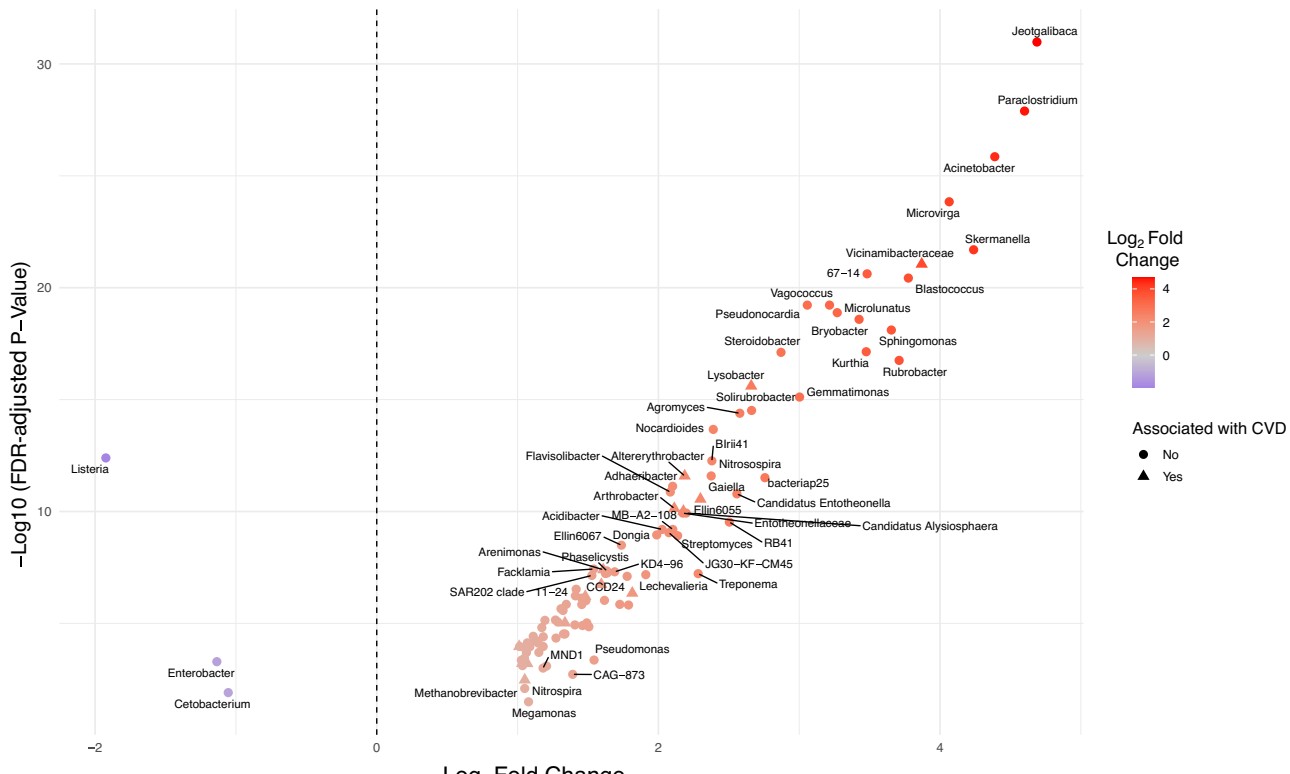

**Fig. 3 | Differential abundance of gut microbial taxa by *S. mansoni*-infection status.** Volcano plot showing the log₂ fold change in abundance (x-axis) versus the −log₁₀ adjusted *p*-value (y-axis) for microbial taxa differentially abundant between *S. mansoni*-infected and -uninfected individuals. Each point represents a microbial taxon. Taxa to the right of the vertical dashed line are enriched in *S. mansoni*-infected individuals, while those to the left are depleted. Points are coloured by their log₂ fold change, with red indicating higher abundance and blue indicating lower abundance in *S. mansoni*-infected. Labels highlight taxa with statistically significant differences (FDR-adjusted *p* < 0.05). Also, coloured taxa denoted by triangular dots are both significantly enriched in *S. mansoni* participants and associated with CVD risk. Those taxa shown by circular dots are significantly impacted by *S. mansoni* infection but are not associated with CVD risk.

discriminant analysis (LDA) using LEfSe identified bacterial genera that best discriminated between *S. m+* and *S. m−* individuals based on relative abundance patterns (Fig. 2D–E). Genera with LDA scores > 2 were considered to have a meaningful effect size in separating the groups. Among rural participants, taxa such as *Acinetobacter, Methanosphaera, Jeotgalibaca*, and *Ruminococcus* were enriched in *S. m+* individuals, while *Streptococcus, Prevotella*, and *Roseburia* were enriched in the *S. m−* group. In the urban cohort, LDA highlighted *Romboutsia, Succinivibrio, Clostridium_sensu_stricto_1, Treponema, Pseudomonas, Butyrivibrio*, and *Gastranaerophilales as* enriched in *S. m + *, whereas *Prevotella, Streptococcus, Dialister, Facalibacterium*, and *Agathobacter* were enriched in *S. m− individuals*. Notably, *Prevotella* and *Streptococcus* were consistently enriched in the *S. m−* group across both settings. To further investigate the impact of *S. mansoni* infection on the gut microbiome, we compared microbial taxonomic abundance profiles between *S. m+* and *S. m*-individuals. Differential abundance analysis revealed a set of taxa significantly enriched in *S. m+* individuals (FDR adjusted *p* < 0.05), including *Altererythrobacter, Arthrobacter Devosia, Domibacillus*, and *Lysobacter* (Fig. 3 and Supplementary Fig. 3). *Listeria, Enterobacter* and *Cetobacterium* were significantly depleted in the *S. m+* individuals.

## Specific microbial taxa mediate the relationship between *S. mansoni* infection and cardiovascular risk

Regression analyses revealed distinct microbe–CVD risk factor associations present in rural (Fig. 4A) and urban (Fig. 4B) sample populations after adjusting for confounding factors including age, sex, BMI, and diet (in the rural population).

In both figures, the microbes shown are the fifty most abundant. Notably, taxa such as *Treponema* were consistently associated with LDL cholesterol in both rural and urban participants.

To further illustrate the degree of overlap and uniqueness of these microbial associations across cardiovascular outcomes, we generated a Venn diagram (Fig. 4C) using a previously published tool[38], which highlights shared and distinct taxa linked to multiple CVD risk factors.

To investigate the functional relevance of the microbial differences reported here, we conducted a mediation analysis to identify taxa that may mediate the impact of *S. mansoni* infection on cardiovascular disease risk factors. Several taxa significantly mediated the relationship between *S. mansoni* infection and reduced cardiovascular risk, with all the microbiota shown in Fig. 4D showing negative effects (p < 0.05) on CVD risk. Among these, *Treponema* mediated reductions in both insulin and glucose levels; *Tabrizicola* contributed to lower systolic blood pressure, LDL cholesterol, total cholesterol, and insulin; *Promicromonospora* mediated reductions in LDL cholesterol; *Papillibacter* mediated lower insulin; *Pedomicrobium* mediated reduced LDL cholesterol; *Catenisphaera* mediated lower total cholesterol; and *CCD24* mediated reductions in LDL cholesterol. Additional infected-enriched taxa, including *Methanobrevibacter, Phoenicibacter, UTCFX1*, and *Roseomonas* mediated decreases in glucose, glucose, systolic blood pressure, and systolic blood pressure respectively. Two taxa were more abundant in uninfected individuals: *Lachnospiraceae_UCG.001* and *Lachnospiraceae_UCG.004*, both of which mediated reductions in systolic blood pressure. Together, these patterns indicate that although most mediating taxa were enriched in infected participants, both infected- and uninfected-abundant microbes

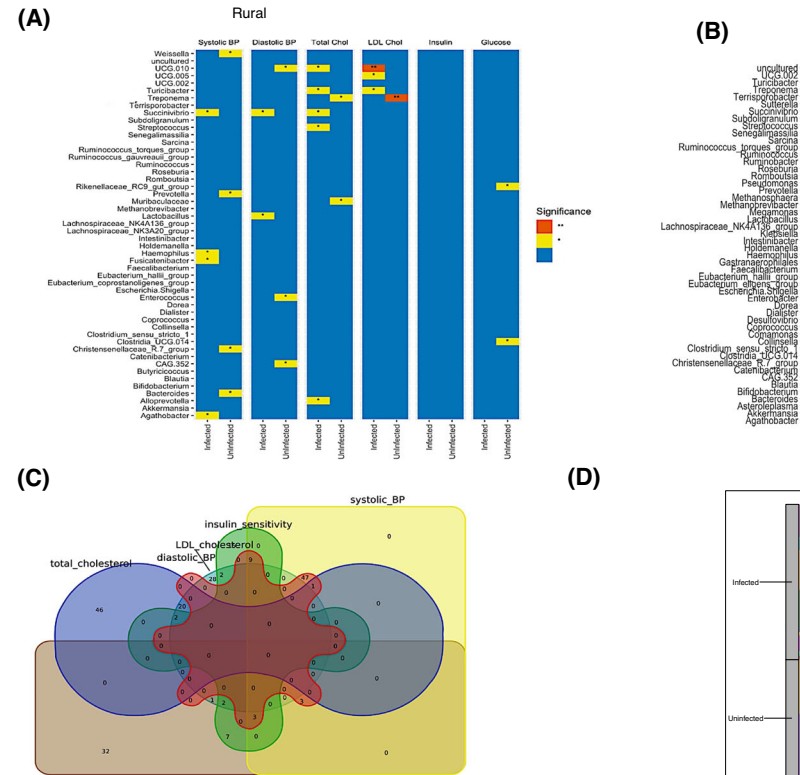

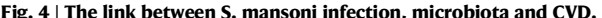

**Fig. 4 | The link between S. mansoni infection, microbiota and CVD.**
**A** Association of microbes and CVD risk in *S. m+* and *S. m-* individuals living in rural setting: The asterisks (*) show the significant association between microbes and the different CVD risk factors in the rural population, following linear regression analysis where potential confounders including age, sex, body mass index (BMI) and diet were adjusted for. These are results from the top 50 abundant microbes. ** indicative of *p* < 0.01, while * indicative of a p-value < 0.05. BP is blood pressure, Total Chol is total cholesterol while LDL Chol is low density lipoprotein cholesterol. Taxa such as *Treponema* were consistently associated with LDL cholesterol in both rural and urban populations in (**B**). **B** Linear regression analysis like what is shown in B but for *S. m+* and *S. m-* individuals living in urban setting. Potential confounders including age, sex and body mass index (BMI) were adjusted for. *** show p. value < 0.001, ** indicative of *p* < 0.01, while * shows *p* value < 0.05. **C** Venn diagram showing overlap among microbial taxa that are significantly associated with

different cardiovascular risk factors. Associations were identified using multivariable linear regression models (adjusting for age, sex, BMI, diet). All hypothesis tests were two-sided, and microbial taxa were considered significant at *P* < 0.05 without adjustment for multiple comparisons. **D** Mediation analysis (adjusted for age, sex and residential setting) illustrating the mediatory role of microbes in *S. mansoni*-associated modulation of CVD risk. An alluvial plot showing microbes through which helminths may alter one's CVD risk. Mediation analysis was performed using non-parametric bootstrap resampling with two-sided hypothesis testing as implemented in the Pingouin Python library. Microbial taxa were first identified as differentially abundant between the *S. m+* and *S. m-* groups, following multiple-testing correction (false discovery rate–adjusted *P* < 0.05), and subsequently showed significant negative mediation effects on the CVD risk factors shown, defined by 95% bootstrap confidence intervals not crossing zero. No additional adjustment for multiple comparisons was applied to the mediation analyses.

exhibited negative mediation effects. Additionally, Enterobacter and Klebsiella had positive mediation effects on total cholesterol and insulin sensitivity respectively, among infected individuals (Supplementary Fig. 4).

### *S. mansoni* infection is associated with metabolome differences
To dissect whether the effect of *S. mansoni* infection on the gut microbiome can translate into differences in microbial-related metabolism, we compared faecal metabolomic profiles between *S. m+* and *S. m-* individuals. A volcano plot shows the differentially abundant (*p* < 0.05) metabolites in both groups, highlighting metabolic alterations associated with *S. mansoni* infection (Fig. 5A). The 10 most enriched metabolites in the infected group include HMDB36635, HMDB39448, HMDB10385, HMDB14867, HMDB08887, HMDB30053, HMDB31040, HMDB60963, HMDB11158 and metabolite with mass to charge ratio 6.26_1326581m/z. The 10 most enriched metabolites in the uninfected group include HMDB46827, HMDB29485, HMDB14388, HMDB36122, HMDB31828, HMDB14377, HMDB10261, HMDB14585, HMDB11367, HMDB11895. Further, partial least squares discriminant analysis (PLS-DA) suggested some degree of separation between the faecal metabolomic profiles of *S. m+* and *S. m-* individuals (Fig. 5B); however, this clustering did not reach

statistical significance based on PERMANOVA (*p* = 0.48). While these observations provide important insights into potential microbiome-mediated pathways, the metabolomic differences should be interpreted as hypothesis-generating rather than constituting a definitive infection-related metabolic signature, particularly given the modest global separation in untargeted metabolomics. To evaluate the discriminative capacity of metabolomic features, we trained a PLS-DA–based classification model (Supplementary Fig. 5). The model exhibited limited predictive performance, with an overall accuracy of 53.8%, specificity of 54.8%, and sensitivity of 53.2%, indicating poor ability to discriminate between *S. m+* and *S. m-* individuals. Further, in Supplementary Fig. 2 we compared metabolomes of participants in the rural setting that were in the intensive anthelminthic treatment arm to those in the standard anthelminthic treatment and we found no difference.

### Enrichment of lipid-related pathways among metabolites elevated in *S. mansoni*-infected individuals
Comparative analysis revealed a distinct metabolic signature in infected individuals, with a subset of metabolites significantly more abundant compared to uninfected controls. Pathway enrichment analysis of these elevated metabolites was performed using the Integrated

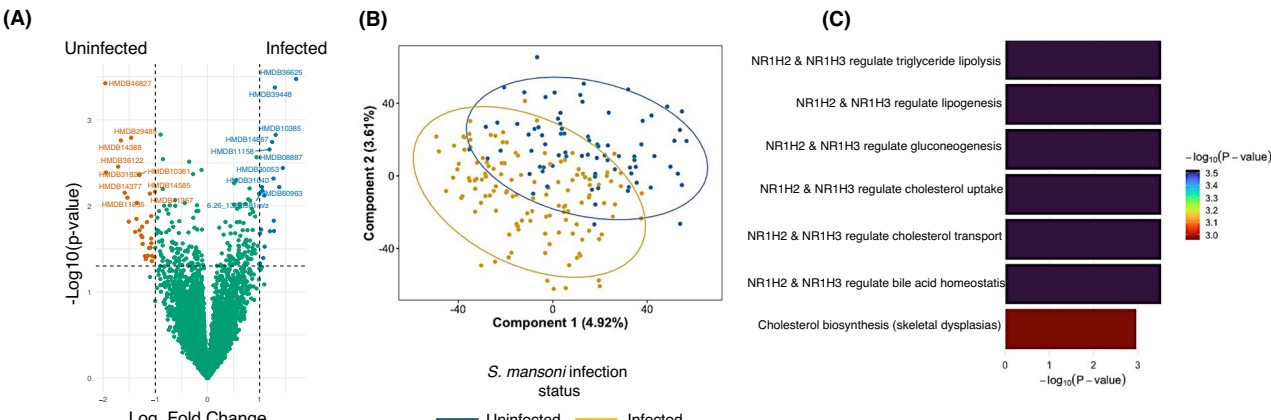

**Fig. 5 | Differential abundance of metabolites by *S. mansoni*-infection status.**
**A** Volcano plot showing metabolites that are differentially abundant between the S. m-infected ($n = 128$) and uninfected groups ($n = 81$). Metabolite-level comparison was performed using two-sided independent-samples t-tests with mean differences and 95% confidence intervals for derived for each metabolite. No adjustment for multiple comparisons was applied. Metabolites with $p \leq 0.05$ and log2 fold change $\geq 1.0$ were significantly more abundant in infected, while metabolites with $p \leq 0.05$ and log2 fold change $\leq 1.0$ were considered more abundant in *S. m*-uninfected group. Red dots at the upper left area are significantly upregulated in the uninfected group, blue dots located at the upper right area are significantly upregulated in *S. m*-infected individuals. The top 10 most differentially abundant metabolites from either group were labelled. **B** Partial least squares-discriminant analysis (PLS-DA) score plot showing differences in clustering of metabolites in *S. m+* and *S. m-* individuals. The blue represents *S. m-* while yellow represents the *S. m+* individuals.

We assessed group-level differences in metabolomic profiles using PERMANOVA on scaled Euclidean distances. *S. m+* vs *S. m-* profiles were not significantly different ($p = 0.48$, $R^2 = 0.00452$, F = 0.9301, Number of permutations= 999) (**C**) Biological pathways enriched among metabolites that were more abundant in *S. m+* compared to *S. m-*individuals. Metabolites were profiled from faecal samples using liquid chromatography–mass spectrometry and identified using Progenesis QI software. Metabolites with a maximum fold change $\geq 1.5$ between groups and two-sided ANOVA $P < 0.05$ were retained for downstream analysis. Pathway enrichment analysis was performed using Integrated Molecular Pathway Level Analysis (IMPaLA). The bar plot shows the $-\log_{10}$-transformed $P$ values for the top enriched biological pathways, with bar colour intensity reflecting the strength of statistical significance. Higher $-\log_{10}(P$ value) indicates stronger enrichment. No adjustment for multiple comparisons was applied at the pathway level. Notably, pathways linked to lipid metabolism and cholesterol regulation were significantly enriched.

**Table 2 | Biological pathways enriched by metabolites that were more abundant in *S. mansoni* infected individuals than uninfected participants**

| Biological pathway enriched by metabolites | Pathway source | Number of metabolites | P- value |
|---|---|---|---|
| NR1H2 & NR1H3 regulate gene expression to limit cholesterol uptake | Reactome | 3 | 0.0003 |
| NR1H2 & NR1H3 regulate gene expression to control bile acid homoeostasis | Reactome | 3 | 0.0003 |
| NR1H2 & NR1H3 regulate gene expression linked to gluconeogenesis | Reactome | 3 | 0.0003 |
| NR1H2 & NR1H3 regulate gene expression linked to lipogenesis | Reactome | 3 | 0.0003 |
| NR1H2 & NR1H3 regulate gene expression linked to triglyceride lipolysis in adipose | Reactome | 3 | 0.0003 |
| NR1H3 & NR1H2 regulate gene expression linked to cholesterol transport and efflux | Reactome | 3 | 0.0003 |
| Cholesterol biosynthesis with skeletal dysplasias | Wikipathways | 4 | 0.0011 |

Metabolites were extracted from faecal samples of 209 participants and profiled using liquid chromatography-mass spectrometry. Metabolites with a maximum fold change ≥ 1.5 between *S. mansoni*–infected (128) and uninfected (81) individuals and two-sided ANOVA $P < 0.05$ were retained for downstream analysis. Pathway analysis was performed using Integrated Molecular Pathway Level Analysis (IMPaLA) with a one-sided hypergeometric test. $P$ values indicate pathway level enrichment significance and are reported as exact values without multiple testing correction. Pathways related to lipid metabolism and cholesterol regulation were among the most significantly enriched.

Molecular Pathway Level Analysis (IMPaLA) platform and results shown in Table 2, and Fig. 5C.

This analysis identified an overrepresentation of pathways regulated by the nuclear receptors NR1H2 (LXRβ) and NR1H3 (LXRα), which are central to lipid homoeostasis and metabolic regulation. Specifically, six Reactome pathways driven by NR1H2/NR1H3 activity were significantly enriched (all $p = 0.0003$), including those regulating cholesterol uptake, bile acid homoeostasis, gluconeogenesis, lipogenesis, triglyceride lipolysis in adipose tissue, and cholesterol transport and efflux. These findings suggest coordinated transcriptional regulation of lipid metabolic processes in *S. mansoni*-infected individuals. In addition, a Wikipathways entry linked to cholesterol biosynthesis in the context of skeletal dysplasias was significantly enriched ($p = 0.0011$). Collectively, these data indicate that *S. mansoni* infection is associated with a specific faecal metabolic profile marked by enhanced abundance of metabolites involved in lipid signalling and transport.

**Integrated microbiome–metabolome interactions link to total and LDL cholesterol levels**
To examine how microbial and metabolic alterations interact to influence lipid metabolism, we conducted integrative correlation analyses, combining microbes and metabolites that were significantly associated with total and LDL cholesterol. Among those microbes and metabolites significantly associated with total cholesterol, a circos plot highlighted robust correlations ($r \geq 0.7$) between specific genera and metabolites (Fig. 6A), which were visualised in detail in a corresponding heatmap (Fig. 6B). Metabolite classification using ClassyFire revealed enrichment of glycerolipids, steroids and steroid derivatives, and glycerolphospholipids among total cholesterol-associated compounds that are linked to the microbes associated with total cholesterol (Fig. 6C).

A similar analysis was done for microbes and metabolites significantly associated with LDL cholesterol and uncovered a distinct but overlapping set of microbe–metabolite correlations ($r \geq 0.65$;

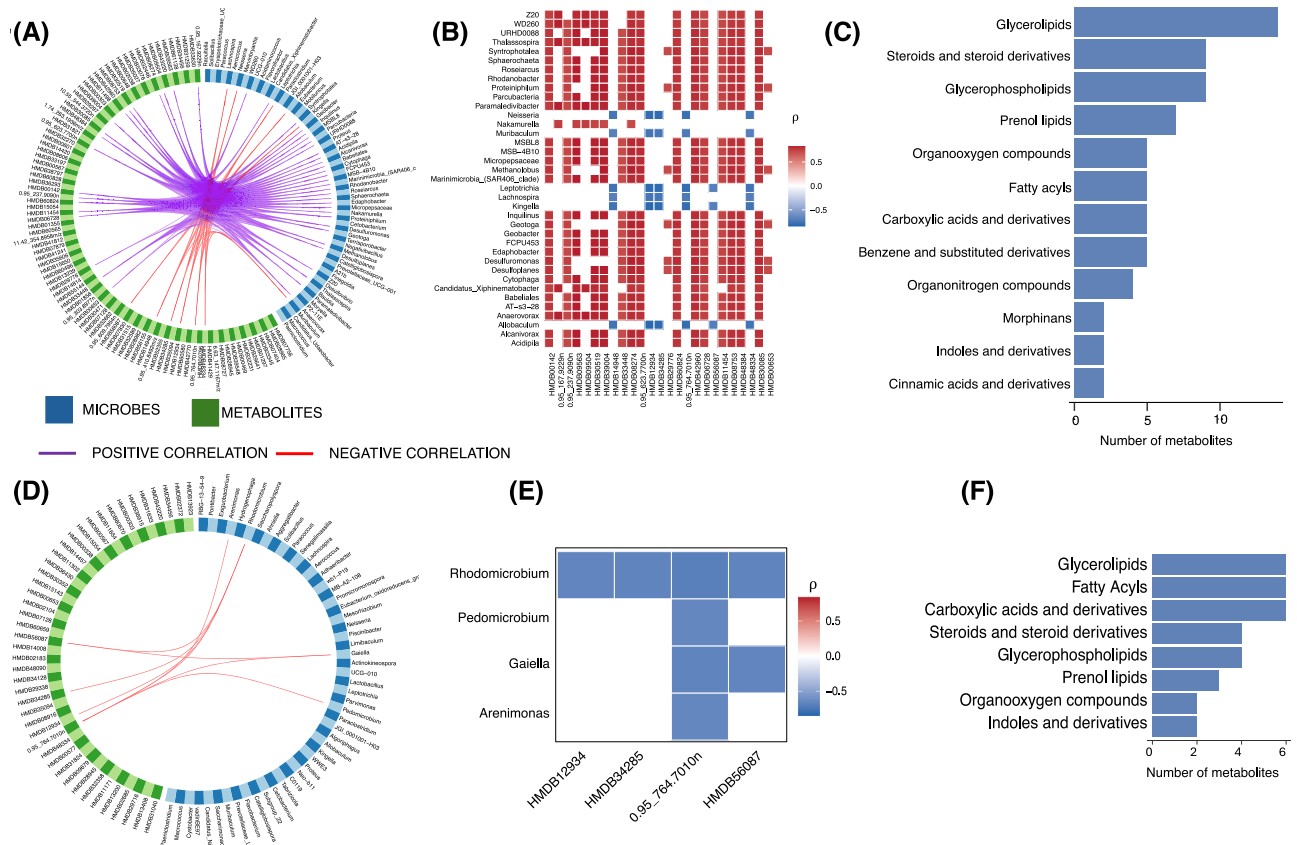

**Fig. 6 | Correlation between cholesterol-associated microbiota and metabolites. A** Circos plot illustrating the strongest correlations (Spearman's r ≥ 0.7) between microbiome (blue blocks) and metabolome (green blocks) features significantly associated with total cholesterol. Associations were identified using models adjusted for age, sex, BMI, and setting as an interaction term (see Fig. 3). Purple lines indicate positive correlations; red lines indicate negative correlations. **B** Heatmap showing the correlation strength between microbial genera and metabolites extracted from panel A. Red intensity reflects positive correlation strength; blue indicates negative correlations. **C** Classification of total cholesterol-associated metabolites using CLASSYFIRE. Horizontal bars represent the number of metabolites per chemical class (y-axis). **D** Circos plot showing top correlations (r ≥ 0.65) between microbiome and metabolome features significantly associated with LDL-cholesterol, using the same model adjustments as in panel A. Green lines indicate positive correlations; red lines indicate negative correlations. **E** Heatmap of LDL-cholesterol-associated microbial genera and metabolites, following the same format and colour scale as (**B**). **F** Classification of LDL-cholesterol-associated metabolites using CLASSYFIRE, as in (**C**).

Fig. 6D–E), again featuring key taxa and metabolite classes previously implicated in lipid homoeostasis (Fig. 6F).

Metabolite annotation revealed that several cholesterol-associated compounds belonged to key chemical classes, including Glycerol lipids, fatty acyls carboxylic acids and derivatives, steroids and steroid derivatives, and glycerolphospholipids (Fig. 6F). There is an overlap in the classes of metabolites linked to total and LDL cholesterol.

### Microbiome–metabolome interactions also relate to blood pressure regulation

We extended our integrative approach to blood pressure phenotypes. Diastolic blood pressure was associated with a network of microbiota–metabolite interactions (r ≥ 0.7; Fig. 7A), and a heatmap visualisation confirming the several strong correlations, both positive and negative as shown in Fig. 7B. Metabolite classification highlighted compounds linked to classes such as glycerolipids, prenol lipids, organooxygen, fatty acyls and steroid and steroid derivatives (Fig. 7C). Microbiome-metabolome associations were found for systolic blood pressure (Fig. 7D–F) and similar classes including prenol lipids, gycerolipids, fatty acyls characterised most of the metabolites involved, emphasizing the role of gut microbial metabolites as potential regulators of blood pressure. Similar analysis was done for insulin associated microbes and metabolites and

fewer microbe-metabolite associations were seen, as shown in supplementary Fig. 6.

### *S. mansoni*-induced microbial changes alter host CVD risk through metabolites

To explore potential mechanistic links between schistosomiasis-associated gut microbiota and risk for CVD, we constructed a directed network integrating differentially abundant microbial taxa, correlated faecal metabolites, and associated CVD risk factors. Among the taxa enriched in schistosomiasis-positive individuals (log$_2$ fold change > 1, FDR-adjusted $p < 0.05$), we identified several genera−including *Lysobacter*, *Arthrobacter*, and *Vicinamibacteraceae*−that were strongly inversely correlated with specific metabolites, such as HMDB31050 and HMDB32627 (|ρ| ≥ 0.65, FDR-adjusted $p < 0.05$), shown in supplementary table 4. These metabolites, in turn, were significantly associated with CVD risk factors, most notably diastolic blood pressure and LDL cholesterol. Visualisation of the network (Fig. 8) revealed a coherent directional path from microbial taxa to metabolite changes and CVD risk, suggesting a putative microbiome−metabolite−CVD axis modulated by *S. mansoni*. These findings support the hypothesis that helminth infection may influence CVD risk through metabolic changes mediated by the gut microbiome. Our findings also showed that *S. mansoni* infection was associated with an enrichment of specific bacterial taxa, including *Domibacillus* and *Gaiella*. Notably, these taxa

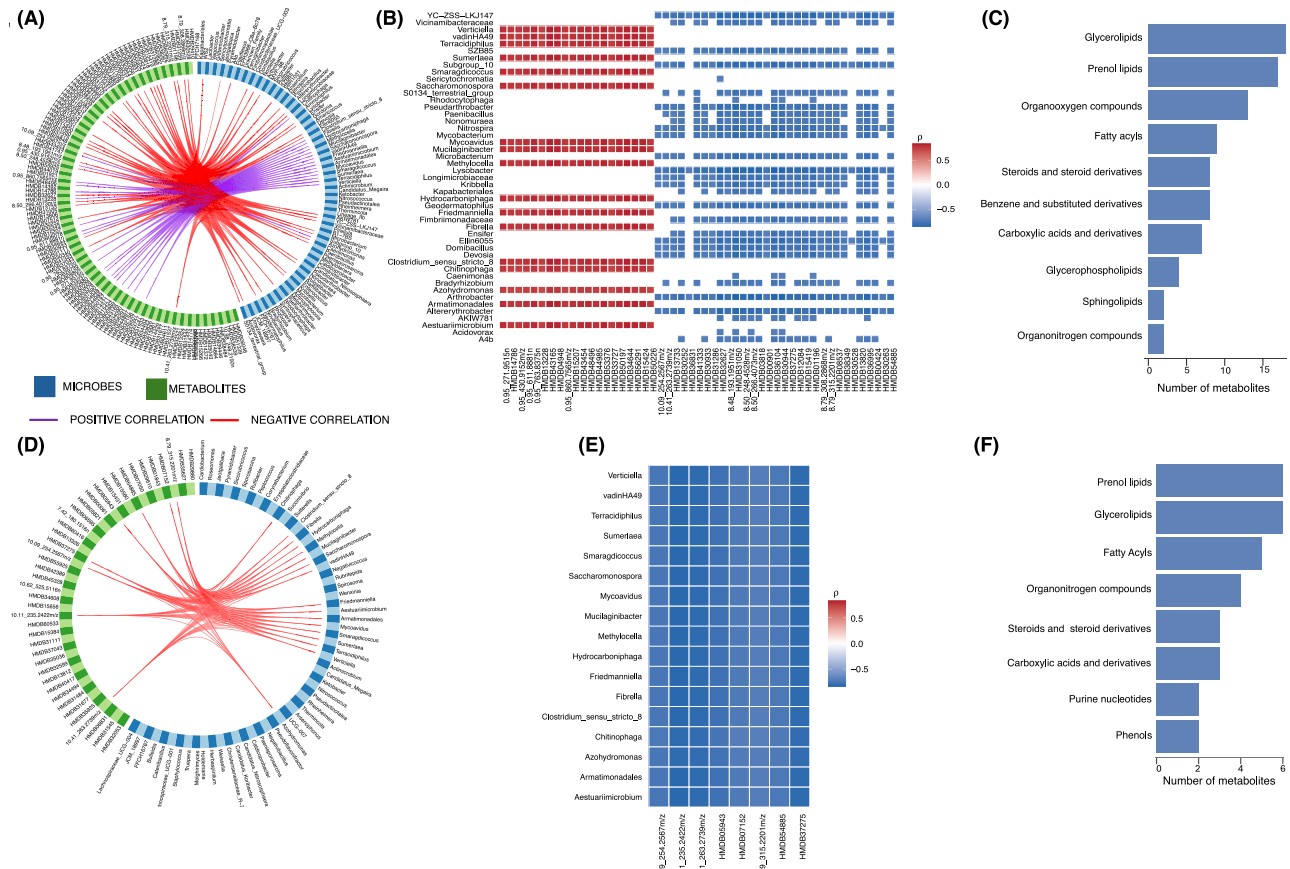

**Fig. 7 | Correlation between blood pressure-associated microbiota and metabolites.** A Circos plot illustrating the strongest correlations (Spearman's r ≥ 0.7) between microbiome (blue blocks) and metabolome (green blocks) features significantly associated with diastolic blood pressure. Associations were identified using models adjusted for age, sex, BMI, and setting as an interaction term (see Fig. 3). Purple lines indicate positive correlations; red lines indicate negative correlations. B Heatmap showing correlation strength between microbial genera and metabolites extracted from panel A. Red intensity reflects positive correlation strength; blue indicates negative correlations. C Classification of diastolic blood pressure-associated metabolites using CLASSYFIRE. Horizontal bars represent the number of metabolites per chemical class (y-axis). D Circos plot showing the strongest correlations (r ≥ 0.70) between microbiome and metabolome features significantly associated with systolic blood pressure, using the same model adjustments as in panel A. Green lines indicate positive correlations; red lines indicate negative correlations. E Heatmap of systolic blood pressure-associated microbial genera and metabolites, following the same format and colour scale as (B). F Classification of systolic blood pressure-associated metabolites using CLASSYFIRE, as in panel C.

exhibited correlations with metabolites and cardiometabolic risk, detailed in Fig. 8.

## Discussion

Our study shows that *S. mansoni* infection is associated with distinct changes in gut microbial diversity, metabolomic profiles, and microbe–metabolic interactions. We can show that these alterations appear to influence cardiovascular risk through multiple, interlinked pathways, implicating the gut ecosystem as a mediator of *S. mansoni*-driven cardiometabolic risk modulation in humans.

The observed differences in alpha diversity between *S. m*+ and *S. m*- individuals suggest that parasitic infection significantly alters one's gut microbial profile. Several studies have reported reduced alpha diversity, typically associated with a less resilient and less functionally diverse microbiome, to be linked to CVD risk. For example, Kelly and colleagues showed an association between increased observed richness and reduced lifetime CVD risk[39]. Similarly, Fu *et al*, reported a positive association between bacterial richness and HDL cholesterol[40] in individuals living in the Netherlands. With such evidence showing that more bacterial diversity and richness is associated reduced CVD risk and improved lipid profiles, therefore we postulated that one way through which *S. mansoni* infection may improve lipid profiles in the host is by increasing bacterial richness and diversity.

In addition to alpha diversity differences, we also observed beta diversity differences in the microbiome profiles between *S. mansoni*-infected and uninfected individuals, further indicating that infection not only increases microbial richness, but it can also shift the overall composition of the microbial community, leading to distinct clustering of infected and uninfected individuals as seen in participants living in urban settings. We did not observe similar differences in clustering of overall microbial structure (beta diversity) between *S. mansoni* infected and uninfected living in the rural setting. This could be because, as shown in previous studies, the rural dwellers tend to have higher gut microbiome diversity and stability due to continuous exposure to a wide range of environmental microbes, diverse diets rich in unprocessed fibre-rich foods, and frequent exposure to infections that may buffer the microbiome against significant changes that may be caused by *S. mansoni* infection[41–44]. Specifically, given the high exposure to *S. mansoni* in our rural population, a typical island community, it is possible that the individuals that were uninfected at the time of sample collection, might have had longstanding effects of *S. mansoni* infection from previous exposure that may modify gut microbiome differences observed in our beta diversity analysis.

Next, we applied linear discriminant analysis (LDA) to identify microbial taxa that best discriminate between individuals with and without *S. mansoni* infection, across both rural and urban settings.

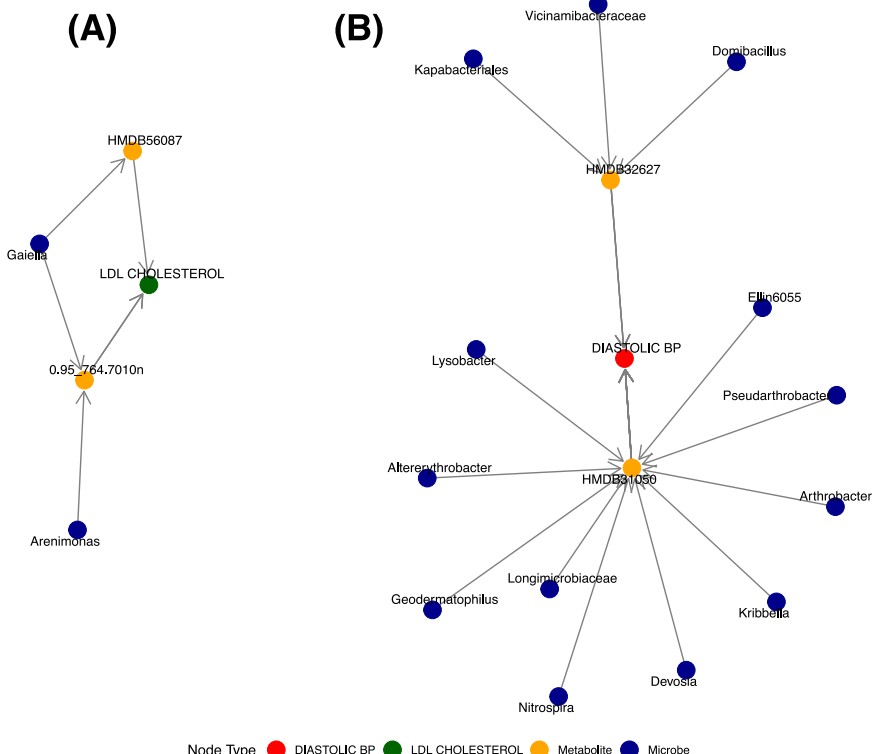

**Fig. 8 | Network representation of microbiome–metabolome–cardiovascular risk in individuals with *S. mansoni*. A** Network showing directional associations between *S. mansoni*–associated gut microbial taxa (dark blue nodes), faecal metabolites (orange nodes), and LDL cholesterol (green node). **B** This network shows directional associations between gut microbial taxa (dark blue nodes), faecal metabolites (orange nodes), and cardiovascular disease Diastolic blood pressure (red node). Arrows indicate significant correlations between nodes, derived from Spearman's correlation analysis. Microbial taxa included in the networks were identified as significantly enriched in *S. mansoni*–infected compared with uninfected individuals using differential abundance analysis, with $\log_2$ fold change > 1 and false discovery rate (FDR)–adjusted $P < 0.05$ (two-sided testing). Associations between microbial taxa and metabolites, and between metabolites and CVD risk factors, were identified Spearman's rank correlation analysis; correlation coefficients ($r$) and exact two-sided $P$ values were calculated for each pairwise association. No additional adjustment for multiple comparisons was applied at the correlation stage. Cardiovascular risk factors are represented by green (LDL cholesterol) and red (diastolic blood pressure) nodes, denoting the specific clinical trait associated with the metabolite and microbe. Edge directionality flows from microbe → metabolite → CVD risk factor, showing a hypothesised mediation path through which *S. mansoni*-associated microbial changes may influence metabolic and cardiovascular profiles.

Unlike statistical tests of differential abundance, which identify taxa that vary significantly in abundance between groups, LDA ranks features based on their ability to separate predefined classes. Notably, microbes such as *Prevotella* and *Streptococcus* were found to be consistently more abundant in *S. mansoni* infected individuals and are known to play pivotal roles in modulating immune responses and inflammation thereby bringing about protection against CVD risk.

These findings align with the hypothesis that parasitic infections such as *S. mansoni* may exert long-term effects on host health by reshaping the microbiome. Importantly, the altered microbial profiles we observed may not only reflect the host's immune response to infection but could also be directly involved in mediating disease risk through metabolic and inflammatory pathways. We therefore needed to investigate the mediatory role that helminth-induced gut microbiota changes could play in altering one's CVD risk.

As such, we performed mediation analysis to show that indeed *S. mansoni* infection may influence cardiovascular risk indirectly through its effects on the gut microbiome. Specifically, we observed that variation in microbial composition was statistically associated with both higher and lower levels of key cardiovascular risk factors, including LDL cholesterol and blood pressure. Several taxa enriched in infected individuals exhibited negative indirect effects, consistent with a pattern that could contribute to the more favourable lipid and blood pressure profiles observed in infected participants. However, these mediation findings reflect associations rather than demonstrated

causal pathways. Although higher microbial richness and infection-related shifts in community structure have been linked to improved metabolic outcomes in prior studies, our data cannot establish that these microbial differences explain or drive the cardiometabolic phenotype. Instead, our findings should be interpreted as identifying plausible microbiome-related pathways that warrant further mechanistic and longitudinal investigation.

Most mediating taxa were more abundant in infected individuals, and several of these contributed to reduced CVD risk through negative indirect effects. These included *Treponema*, *Tabrizicola*, *Promicromonospora*, *Pedomicrobium*, *Papillibacter*, *Catenisphaera*, which variously mediated improvements in LDL cholesterol, systolic blood pressure, glucose, or insulin levels. These findings underscore the complexity of microbiota–host interactions, indicating that the health impact of a given microbe is not solely determined by its presence or absence, but by its context within the broader microbial community and host environment. This functional diversity reinforces the idea that *S. mansoni*-associated shifts in microbiome composition may tip the balance of microbial activity toward either protective or deleterious effects, depending on the taxa involved and the pathways engaged. Importantly, these mediation patterns do not indicate that *S. mansoni* infection confers a uniformly protective metabolic profile; rather, they reveal a mixture of positive and inverse microbial associations that Households were excluded only if all members were absent or declined participation are highly context dependent and should be interpreted

cautiously given the cross-sectional design. Several human and animal studies show that chronic infection can contribute to CVD and hypertension, driven by inflammation around parasite eggs and longer-term changes in vascular structure[45–47]. These observations highlight that helminth infections can exert harmful cardiovascular effects, even as they may show more favourable metabolic associations in some contexts.

Additionally, positive mediation of microbiota on CVD risk factors such as total cholesterol in uninfected individuals shown in Supplementary Fig. 4 could imply that *S. mansoni* infected individuals have less total cholesterol because they lack microbial populations that have been shown to lead to increases in these CVD risk factors.

For example, *Enterobacter* mediated increased total cholesterol in individuals without *S. mansoni* infection. *Enterobacter* is linked with systemic inflammation through lipopolysaccharide (LPS)-mediated activation of host immune pathways[48–50], hence altering lipid metabolism. These findings point to divergent microbial contributions to metabolic and cardiovascular phenotypes depending on infection status. The dual nature of microbial associations emphasises the need to consider ecological context and host-microbe interactions in interpreting microbiome-mediated health outcomes. Overall, these results suggest that *S. mansoni*-associated shifts in the gut microbiome may actively contribute to the modulation of cardiovascular risk factors and offer candidate microbial targets for further mechanistic and translational investigation.

To complement this analysis, we further employed linear regression to investigate the relationship between microbes and CVD risk factors. By adjusting for key confounders such as age, sex, BMI, and diet, we aimed to isolate the unique contribution of the microbes to CVD risk. The findings revealed significant associations between specific microbial taxa and distinct CVD risk factors, suggesting that these microbes may play a mechanistic role in CVD risk.

These results agree with the mediation analysis findings, where the role of *S. mansoni* infection in modulating CVD risk factors was partially explained by its influence on the microbiome. This suggests a pathway wherein microbes, through their microbial composition, may mediate the relationship between S. mansoni and CVD outcomes. The robustness of these associations, even after adjusting for confounders, underscores the potential of these microbial markers as indicators or contributors to CVD disease pathways. These findings support the notion that *S. mansoni* infection and the gut microbiota can influence cardiovascular disease (CVD) risk both independently and concertedly. The identification of taxa that significantly mediate the relationship between *S. mansoni* infection and reduced CVD risk suggests that part of the protective effect of infection may be exerted through infection-induced remodelling of the microbiota–metabolome axis. However, not all associations between *S. mansoni* and cardiovascular risk were microbiota-mediated, indicating the presence of parallel, microbiota-independent pathways such as immune modulation through which helminth infection may confer CVD risk protection.

We observed more significant associations of microbes with CVD risk among *S. mansoni* infected individuals in the urban compared to those in rural populations. The observed stronger associations in the urban population can be attributed to *S. mansoni* infection having a greater influence on microbial diversity in this population compared to the rural. Having shown that even from the top 50 abundant microbes, we have evidence of some microbes being associated with CVD risk, we then extracted all the significantly associated microbes from the entire dataset (beyond 50 most abundant microbes) and investigated if there was any relationship (overlap) between microbes that are significantly associated with the different CVD risk factors. Indeed, we can show there are several microbes that are associated with more than one cardiovascular risk.

Triangulating evidence from association and mediation analysis supports the hypothesis that the gut microbiome could play a central role in the regulation of metabolic pathways that are critical to cardiovascular health. For instance, certain microbial taxa produce metabolites such as SCFAs and secondary bile acids that can influence lipid metabolism[33]. Dysregulation of these pathways may therefore contribute to an increased risk of cardiovascular events. These insights highlight the importance of considering infectious diseases like *S. mansoni* not only in terms of acute morbidity but also in their potential to influence long-term health outcomes through microbiome-mediated mechanisms.

We therefore set out to investigate the metabolome profiles of our participants as a way of assessing if, similar to the microbiome changes observed here, there could be differences in the faecal metabolites between individuals that were infected with *S. mansoni* infection and those that are not infected. This would enable us to infer functionality of the gut microbiota and how they act to alter one's cardiovascular risk.

Despite no statistically significant global separation of metabolomic profiles between *S. mansoni*–infected and uninfected individuals as assessed by PLS-DA, univariate linear regression analysis identified several metabolites that were differentially abundant between the two groups, with multiple features reaching nominal significance thresholds. This suggests that *S. mansoni* infection is associated with specific metabolic alterations rather than broad-scale shifts in the overall metabolome. The lack of clear clustering in multivariate space may reflect substantial inter-individual heterogeneity, or localised metabolic effects of infection, or the multifactorial nature of host metabolic responses. Although we identified differentially abundant metabolites between *S. mansoni*–infected and uninfected individuals, our PLS-DA classification model demonstrated limited predictive performance, likely reflecting the biology of *S. mansoni* transmission. Unlike enteric pathogens whose acquisition may be modulated by gut microbial or metabolic environments, *S. mansoni* is acquired via percutaneous exposure to cercariae-contaminated freshwater. As such, microbiome and metabolome features are unlikely to serve as determinants of infection status. Instead, the observed metabolic shifts are more plausibly consequences of infection, supporting our interpretation that *S. mansoni* may exert causal effects on host physiology.

Our findings from the pathway enrichment analysis revealed that *S. mansoni* infection is associated with distinct alterations in host lipid metabolism, as evidenced by significant enrichment of NR1H2/NR1H3-regulated pathways among the differentially abundant metabolites. These nuclear receptors (LXRα and LXRβ) are key transcriptional regulators of lipid homoeostasis, and their coordinated activation suggests a host response aimed at modulating cholesterol uptake, bile acid turnover, and lipid mobilization during infection. The parallel enrichment of pathways linked to gluconeogenesis and lipogenesis further supports a broader metabolic reprogramming, potentially reflecting shifts in host energy utilisation and storage under chronic parasitic infection.

These findings align with emerging evidence that helminth infections exert broad systemic effects on host physiology, including lipid and glucose metabolism, and may influence susceptibility to non-communicable diseases such as diabetes and cardiovascular disease[27]. Future work should investigate whether modulation of LXR signalling contributes to immune tolerance, pathogen persistence, or protection from metabolic disease in endemic populations.

Beyond pathways, we performed linear regression analysis to identify the metabolites that were significantly associated with the different CVD risk factors and whether there was overlap between these metabolites. We indeed show here that there are metabolites associated with CVD risk and that there is overlap between metabolites that are associated with various CVD risk factors.

With a possibility that the microbes found to be associated with the particular risk factors could correlate with metabolites that are similarly associated with the same CVD risk, we integrated these two

data modalities- microbiome and metabolome. We identified specific microbiome-metabolome signatures that are associated with cardiovascular risk factors including total- and LDL-cholesterol, DBP and SBP. This integrative approach allows us to capture the complex interactions between the gut microbiome and metabolome, revealing, for each CVD risk factor, how shifts in microbial composition may drive changes in metabolite levels.

After identifying the significantly CVD associated microbes and the metabolites that interact together, we further complemented this data with the differential abundance analysis to highlight whether some these CVD-associated microbes interacting with CVD-associated metabolites were enriched in *S. mansoni* infected individuals. By doing so, this allowed us to generate a Schistosomiasis-microbe-metabolite-CVD risk mechanistic network. Our integrative network analysis reveals that several bacterial taxa enriched in *S. mansoni* individuals were strongly correlated with faecal metabolites. These metabolites were, in turn, associated with key CVD risk factors, including diastolic blood pressure and LDL cholesterol.

This directional pattern supports the hypothesis that helminth-induced alterations in the gut microbiota may influence host cardiovascular risk through downstream effects on the metabolome. The inverse associations between these microbes and adverse CVD phenotypes align with emerging evidence suggesting that certain helminth-driven microbial shifts may exert systemic immunometabolic benefits. Notably, taxa such as *Lysobacter* and *Devosia* have been implicated in anti-inflammatory metabolic pathways[51,52], providing a plausible biological basis for these associations. While causality cannot be inferred from this cross-sectional design, these findings raise the possibility that *S. mansoni*, or its microbiome-mediated effects, may modulate cardiometabolic outcomes in endemic settings. Further longitudinal and interventional studies are warranted to validate these interactions and assess their relevance for biomarker development or CVD therapeutic modulation.

In as much as this study provides important insights into the gut microbiome-metabolome-CVD risk interaction in a typical setting with high *S. mansoni* infestation, there are potential limitations. Firstly, we did not comprehensively profile participants' dietary habits, which are known to have profound effects on both the microbiome and metabolome and are likely to represent an unmeasured confounder. Given the variability in diet across different regions and individuals, future studies should incorporate detailed dietary assessments such as next generation sequencing methodologies to disentangle the effects of infection from those of diet.

Secondly, antibiotic exposure represents an important but incompletely measured source of potential variability in microbiome studies. In these Ugandan settings, antibiotics are generally accessible without prescription and can be obtained through informal outlets, making accurate self-report challenging and frequently unreliable. In the rural cohort, limited self-reported medication data were collected as part of the survey. Within these constraints, reported antibiotic use was uncommon, with only six participants reporting use, and other medication categories similarly rare (Supplementary Table 1). Comparable medication data were not available for the urban cohort, precluding meaningful statistical adjustment.

Our analytical strategy was guided by a directed acyclic graph, which identified age, sex, and setting as the minimal sufficient adjustment set required to block backdoor paths between *S. mansoni* infection, the gut microbiome, and cardiometabolic risk factors. While antibiotic exposure is conceptually important, it could not be empirically incorporated into the adjustment set given the absence of systematic and harmonised data, and exclusion or adjustment based on incomplete or asymmetrically measured information would introduce bias without improving causal interpretability. We therefore acknowledge the absence of detailed antibiotic-use data as a limitation and highlight prospective, systematic assessment of antibiotic exposure as a priority for future studies. Furthermore, given the cross-sectional design of our study, we cannot establish causality in the observed associations between *S. mansoni* infection, microbiome composition, metabolite profiles, and cardiovascular risk. Temporal dynamics of microbial and metabolic alterations following infection remain unexplored, limiting our ability to infer directionality. As such, longitudinal studies, ideally spanning pre-infection, active infection, and post-treatment phases, would be essential to disentangle cause-effect relationships and capture the evolving host–microbiome–metabolome interactions over time. Incorporating repeated sampling, coupled with temporal metadata such as infection history and treatment timing, would provide a more robust framework for understanding how *S. mansoni* shapes cardiometabolic risk through microbial and metabolic pathways.

Our findings provide evidence that *S. mansoni* infection is associated with significant alterations in the gut microbiome and metabolome profiles, with important implications for cardiovascular risk. These microbiome-mediated effects may represent a novel pathway through which parasitic infections influence CVD risk. Future studies should focus on refining our understanding of these interactions, with an emphasis on diet, antibiotic use, and circadian regulation of microbial activity. Our study paves the way for developing microbiome-targeted interventions that that may mimic helminth infections to reduce cardiovascular risk in humans, but such approaches will require rigorous mechanistic and interventional validation.

## Methods

This research was conducted in accordance with all relevant ethical regulations. Ethical approval for the parent studies and for the present secondary analyses was obtained from the Uganda Virus Research Institute Research Ethics Committee (UVRI-REC), the London School of Hygiene and Tropical Medicine (LSHTM) Ethics Committee, the Uganda National Council for Science and Technology (UNCST), and the Higher Degrees Research and Ethics Committee of the School of Medicine, College of Health Sciences, Makerere University. Written informed consent was obtained from all participants prior to enrolment in the parent studies, including explicit consent for the storage and future use of biological samples and associated data in secondary analyses such as the present work.

### Study design and sample size considerations

This study is an observational, cross-sectional secondary analysis nested within two previously established cohort studies. No formal statistical power calculation was performed to predetermine sample size for the present analyses.

Instead, sample size considerations were informed by prior microbiome literature indicating that fewer than 50% of operational taxonomic units (OTUs) are typically detectable across faecal samples[40]. Under this conservative assumption, anticipated rural (approximately 120 participants) and urban (approximately 80 participants) sample sizes were expected to yield sufficient samples with detectable microbial features to support multivariable regression modelling and rural–urban comparisons.

This sample size was considered sufficient to support linear regression models including approximately six covariates relevant to cardiometabolic outcomes (for example, age, sex, body mass index, dietary factors, and infection status), while maintaining stable regression coefficients and minimising the risk of model overfitting.

Microbiome sequencing was attempted for all eligible stool samples. Following laboratory and bioinformatic quality control, samples meeting predefined criteria were retained for analysis. The final analytic sample (N = 209) exceeded anticipated minimum thresholds and is reported in the Abstract and Results.

Firstly, we established a comprehensive framework to investigate how *Schistosoma mansoni* infection influences cardiovascular disease

risk through alterations in the gut microbiome and metabolome (Fig. 1).

We used samples from rural participants in the LaVIISWA trial[27,53], and a second, well-characterised survey in a nearby urban setting in Uganda[54].

LaVIISWA was a cluster-randomised trial conducted among Lake Victoria Island fishing communities in Mukono district, Uganda. The study was conducted in 27 fishing villages: one was selected for piloting the study and the remaining 26 were randomised in a 1:1 ratio to standard deworming (single dose praziquantel given once a year and single dose albendazole twice a year) or intensive deworming (single dose praziquantel and triple dose albendazole four times a year). The samples we used were collected during the metabolic survey undertaken after 4 years of intervention. Contemporaneously, the rest of the samples for our study were collected from participants of an Urban Survey that was conducted in Entebbe municipality (an urban setting) found on shores north of Lake Victoria. Entebbe is in Wakiso district approximately 40 km southwest of Kampala, the Ugandan capital city. In the urban survey, no community-wide anthelminthic treatment programme was implemented, and participants therefore had no treatment assignment. We have previously reported that the rural population showed a markedly higher burden of helminth infections, with *Schistosoma mansoni* detected significantly more frequently than in the urban group, as illustrated by both stool Kato-Katz microscopy (31.7% vs 9.9%, $p < 0.001$) and stool PCR analysis (47.6% vs 22.2%, $p < 0.001$)[54].

In both studies, following overnight fasting, stool and blood samples were collected from the participants. Metabolic outcomes measured are: fasting blood sugar, insulin levels, serum lipid levels, body mass index (BMI), waist and hip circumference and blood pressure (systolic and diastolic).

## Exposure and outcome assessment

**Sociodemographic and anthropometric data.** Age and sex were documented using a structured survey tool. Body weight was recorded to the nearest 0.1 kg using a digital scale (SECA model 875), with participants lightly clothed and barefoot. Standing height was measured to the nearest 0.1 cm using a portable stadiometer (SECA model 213). Waist circumference was measured midway between the lower rib and iliac crest, while hip circumference was taken at the level of the greater trochanters, both using a non-elastic measuring tape. Body mass index (BMI) was calculated as weight (kg) divided by height (m²), and waist-to-hip ratio was derived accordingly.

## Parasitological assessment

Stool samples were analysed for helminth infections using both microscopy and molecular techniques. The Kato-Katz method was used to quantify *S. mansoni* and *T. trichiura* eggs as described by Sanya et al.[27]. Real-time PCR assays were employed to detect DNA of *S. mansoni*, hookworm (*N. americanus*), and *S. stercoralis*, the latter being exclusively detected by PCR. PCR data were prioritised for diagnostic confirmation of hookworm due to slide timing variability.

## Dietary Intake

Dietary patterns were evaluated using a semi-quantitative food frequency questionnaire (FFQ) tailored to reflect local Ugandan food consumption habits. The FFQ assessed usual intake in a typical week, covering major food categories such as cereals, legumes, animal proteins, dairy, fruits, vegetables, oils, and sugary drinks. Responses were used to compute dietary diversity scores to adjust for potential confounding in downstream microbiome and metabolome analyses.

## Blood pressure

Blood pressure was measured three times at five-minute intervals in a seated, rested position using a validated automatic sphygmomanometer (OMRON M2, HEM-7121-E), with the average of the last two readings used for analysis. Blood pressure monitors were routinely calibrated through the Uganda National Bureau of Standards to ensure accuracy.

## Cardiometabolic biomarkers

Fasting venous blood samples were collected into EDTA, fluoride oxalate, and serum separator tubes following an overnight fast ($\geq 8$ hours). Participants were advised to avoid physical exertion and tobacco use prior to sampling. Plasma and serum were separated within one hour of collection and cryopreserved in liquid nitrogen.

All biochemical analyses were performed at the MRC/UVRI & LSHTM Uganda Research Unit laboratory (Entebbe) using a Roche Cobas 6000 platform (c 501 module). Fasting plasma glucose and serum lipids—total cholesterol, triglycerides, HDL-c, and LDL-c were quantified using enzymatic colorimetric methods. HbA1c was assessed in whole blood using a turbidimetric inhibition immunoassay, and fasting insulin via electrochemiluminescence immunoassay (ECLIA). Insulin resistance was calculated using the Homoeostasis Model Assessment (HOMA-IR)[55].

## Eligibility criteria

Participants included in this study were drawn from two established cohort studies in Uganda: the LaVIISWA trial (rural) and the Entebbe Urban Survey (urban). Full recruitment procedures for these parent cohorts have been described previously. Briefly, both studies conducted household-based sampling, with households eligible if at least one adult resident provided consent. Households were excluded only if all members were absent or declined participation.

For the present study, we applied additional individual-level inclusion and exclusion criteria to ensure harmonised infection, microbiome, and cardiometabolic datasets across the rural and urban sites. Individuals were included if *Schistosoma mansoni* infection status could be determined using our harmonised diagnostic panel comprising Kato–Katz microscopy and PCR assays. Eligible participants were also required to have complete cardiovascular risk factor data, including systolic and diastolic blood pressure, fasting glucose and insulin, and both total and LDL cholesterol. A further requirement was the availability of a stored stool sample of adequate quality for 16S rRNA gene sequencing, along with complete information for core covariates—age, sex, body mass index, and residential setting. Only individuals who had provided informed consent within the parent studies that included permission for secondary analyses of biological samples were eligible. No additional exclusions were applied beyond the criteria described above.

## Stool sample collection and processing

Stool samples were collected under standardised conditions to ensure preservation of microbial community composition and DNA integrity. In the rural setting, stool samples were obtained during the metabolic outcomes survey of the LaVIISWA, conducted between April and November 2017. In the urban setting, samples were collected in Entebbe municipality, Wakiso district, as part of a deliberately parallel survey conducted between September 2016 and September 2017, designed to enable direct rural–urban comparison of metabolic and immunological outcomes. Each participant was provided with a sterile, screw-capped stool collection container and instructed on hygienic self-collection on the morning of the scheduled visit. Upon arrival at the field site, a portion of each sample was immediately processed for *Schistosoma mansoni* detection using the Kato–Katz method. The remaining stool was transferred using asterile wooden spatulas into pre-labelled cryovials containing 95% molecular-grade ethanol to stabilise microbial DNA and minimize compositional changes during handling.

All ethanol-fixed samples were promptly placed in liquid-nitrogen charged dry shipper in the field (typically within one hour of collection) and maintained at cryogenic temperatures throughout transport to the central laboratory. On arrival, samples were transferred to −80 °C freezers for long-term storage until DNA extraction. All field and laboratory personnel were trained in biospecimen handling, and procedures were performed under aseptic conditions according to harmonised quality control protocols. This workflow ensured uniformity of pre-analytical processing and high confidence in the validity of downstream microbiome measurements. All samples were processed and stored following identical protocols across sites; no batch-specific storage procedures were used.

### Microbiome profiling

Selected samples were prepared from MRC/UVRI and LSHTM-Uganda Research Unit and shipped to Novogene for 16S *rRNA* sequencing. From stool samples collected from these participants at the time the CVD measurements were done, we profiled gut microbial diversity and performed untargeted metabolomics. Genomic microbial DNA was extracted from 150 mg of faecal sample of every selected individual, using the QIAamp DNA Stool kits (Qiagen, Hilden, Germany) according to the manufacturer's instructions. Amplicon-based 16S rRNA gene sequencing targeting the V3–V4 hypervariable regions was performed using a paired-end Illumina sequencing platform.

Following extraction, DNA concentration and purity were assessed using a NanoDrop 2000 spectrophotometer (Thermo Fisher Scientific), and integrity was evaluated via 2% agarose gel electrophoresis. DNA samples (5 μL) were mixed with 1 μL of 6× loading dye and loaded alongside a 1 kb DNA ladder (Thermo Scientific) into a 2% agarose gel prepared with Tris-Acetate-EDTA (1 xTAE) buffer and stained with ethidium bromide (0.5 μg/mL). Electrophoresis was performed at 100 volts for approximately 45 minutes. Gels were visualised using a UV transilluminator, and high molecular weight DNA was confirmed by the presence of a distinct, unsmeared band above 10 kb. Only samples with high-quality, intact DNA were retained for downstream amplification and sequencing.

The 16S library preparation protocol (Reference No: GHFS-LH-039) from Institute of Food Research was used to amplify the V3-V4 hypervariable regions of the bacterial 16S rRNA genes to profile the gut microbiota. The same amount of PCR products from each sample was pooled, end-repaired, A-tailed and further ligated with Illumina adapters. Libraries were sequenced on a paired-end Illumina platform to generate 250 bp paired-end raw reads.

The library was checked with Qubit and real-time PCR for quantification and bioanalyzer for size distribution detection. Quantified libraries were pooled and sequenced on Illumina platforms, according to effective library concentration and data amount required. Paired-end reads were assigned to samples based on their unique barcodes and were truncated by cutting off the barcodes and primer sequences. Paired-end reads were merged using FLASH (Version 1.2.11)[56], a fast and accurate analysis tool designed to merge paired-end reads when at least some of the reads overlap with the reads generated from the opposite end of the same DNA fragment, and the splicing sequences were called Raw Tags. Quality filtering on the raw tags was performed using the fastp (Version 0.20.0) software to obtain high-quality Clean Tags. The Clean Tags were compared with the reference database (Silva database https://www.arb-silva.de) using Vsearch (Version 2.15.0) to detect the chimera sequences, and then the chimera sequences were removed to obtain the EffectiveTags[57].

For the Effective Tags obtained previously, denoise was performed with DADA2 to obtain initial Amplicon Sequence Variants (ASVs) and then ASVs with abundance less than 5 were filtered out[58,59]. Species annotation was performed using QIIME2 (v2023.2) software[60]. The annotation database used was Silva Database. To study phylogenetic relationship of each ASV and the differences of the dominant

species among different samples(groups), multiple sequence alignment was performed using QIIME2 software. The absolute abundance of ASVs was normalised using a standard of sequence number corresponding to the sample with the least sequences. Subsequent analyses of alpha diversity and beta diversity were performed based on the output normalised data.

The rarefaction curves (cutoff = 82,695 reads) for all samples rapidly approached a plateau, indicating that the captured sequencing depth was adequate and that additional reads would not meaningfully increase observed alpha diversity (Supplementary Fig. 7). Likewise, the species accumulation analysis, based on >10 samples as recommended, showed stable asymptotes in species richness, confirming that both sequencing depth and sample size were sufficient to characterise microbial community structure.

### Controls, contamination mitigation, and replication

No mock community (positive control) samples were included. Negative extraction controls and PCR controls were processed alongside biological samples to monitor for reagent and laboratory contamination. Stool samples represent a high-biomass specimen type; nevertheless, potential low-biomass contamination was mitigated through the inclusion of negative controls and downstream computational filtering during bioinformatic processing. No systematic contamination was detected in negative controls.

No biological or technical replicates were sequenced. Each participant contributed a single stool sample that was processed once through DNA extraction, amplification, and sequencing.

### Metabolite extraction and profiling from stool samples

Metabolites were extracted from stool samples using a solid-phase extraction (SPE) approach. Briefly, 1.25 mL of 80:20 methanol:water solution was added to each faecal sample, followed by vortexing and addition of 1 mL of the resulting mixture to 9 mL of molecular-grade water in a 15 mL conical tube. The mixture was centrifuged at 1735 g for 1 minute, and 5 mL of the resulting supernatant was loaded onto SPE cartridges via a syringe. The cartridges were dried by pushing air through them twice using the same syringe and then sealed for shipment to the analytical laboratory in Manchester. A blank control sample without faecal material was prepared alongside the experimental samples.

Upon receipt, metabolite elution was performed using 1.5 mL of 85:15 acetonitrile:methanol, drawn into a 2 mL luer-lock syringe and passed through the cartridge into a sterile 1.5 mL microcentrifuge tube. After allowing 1 minute of solvent equilibration to re-solvate the stationary phase, the eluate was collected at a rate of approximately one drop per second. Samples were then subjected to nitrogen blowdown drying in batches of up to 50, using a 60-position dryer platform.

Dried samples as provided were resuspended in 100 μl 5:95 acetonitrile/water and centrifuged at 20,000 x *g* for 3 min. The top 80 μl supernatant was transferred to a glass autosampler vial with 300 μl insert and capped. Quality control samples were made by pooling 5 μl from each sample.

Liquid chromatography-mass spectrometry analysis was performed using a Thermo-Fisher Ultimate 3000 HPLC system consisting of an HPG-3400RS high pressure gradient pump, TCC 3000 SD column compartment and WPS 3000 Autosampler, coupled to a SCIEX 6600 TripleTOF Q-TOF mass spectrometer with TurboV ion source. The system was controlled by SCIEX Analyst 1.7.1, DCMS Link and Chromeleon Xpress software.

A sample volume of 5 μL was injected by pulled loop onto a 5 μL sample loop with 150 μl post-injection needle wash with 5:95 acetonitrile and water. Injection cycle time was 1 minute per sample. Separations were performed using a Thermo Accucore C18 column with dimensions of 150 mm length, 2.1 mm diameter and 2.6 μm particle size equipped with a guard column of the same phase. Mobile phase A

was water with 0.1 % formic acid; mobile phase B was acetonitrile with 0.1 % formic acid. Separation was performed by gradient chromatography at a flow rate of 0.3 ml/min, starting at 5 % B for 1 minute, ramping to 100 % B over 7 minutes, hold at 100 % B for 2 minutes, then back to 5 % B. Re-equilibration time was 4 min. Total run time including 1 minute injection cycle was 15 minutes.

The mass spectrometer was run in positive mode under the following source conditions: curtain gas pressure, 50 psi; ionspray voltage, 5500 V; temperature, 400 °C; ESI nebuliser gas pressure, 50 psi; heater gas pressure, −70 psi; declustering potential, −80 V.

Data were acquired in an information dependent manner across 10 high sensitivity product ion scans, each with an accumulation time of 100 ms and a TOF survey scan with accumulation time of 250 ms. Total cycle time was 1.3 s. Collision energy was determined using the formula $CE (V) = 0.084 \times m/z + 12$ up to a maximum of 55 V. Isotopes within 4 Da were excluded from the scan.

Acquired data were checked in PeakView 2.2 and imported into Progenesis Qi 2.4 for metabolomics, where they were aligned, peaks were picked, normalised to all compounds and deconvoluted according to standard Progenesis workflows. Signal normalisation was performed using Progenesis QI's default global scaling approach, whereby feature intensities were normalised to the total abundance of all detected compounds per sample. Blank controls were used to assess background contamination introduced during extraction and analysis, while pooled quality control samples were used to monitor retention time alignment, signal reproducibility and analytical stability across the run.

Annotations were made by searching the accurate mass, MS/MS spectrum and isotope distribution ratios of acquired data against the NIST MS/MS metabolite library. Metabolites were identified by searching retention times and accurate masses against an in-house chemical standard library. A validated identification is only given if identical hits are made against both the NIST MS/MS and in-house chemical standard libraries.

## Statistical and computational analysis

**Microbiome data analysis.** To analyze the diversity, richness and uniformity of the communities in the sample, alpha diversity was calculated from indices, including Shannon, observed richness and Pielou_e. Statistical comparisons between infected and uninfected groups were performed using the Krusal-Wallis test (two sided).

Beta diversity was evaluated using Bray−Curtis dissimilarity to compare community structure between samples. Principal coordinates analysis (PCoA) was used to visualise ordination, and PERMANOVA (Adonis) implemented in the 'vegan' R package, with 999 permutations, was used to assess statistical differences in beta diversity across infection groups.

*Differential abundance analysis* To identify microbial taxa differentially abundant between *Schistosoma mansoni*−infected and uninfected individuals, we performed differential abundance (DA) testing using the DESeq2 method implemented within the phyloseq R package[61]. A Wald test was applied to estimate $\log_2$ fold changes in microbial abundance between the two groups. Resulting *p*-values were adjusted for multiple testing using the Benjamini−Hochberg false discovery rate (FDR) procedure. Taxa with an FDR-adjusted *p* < 0.05 were considered statistically significant. A volcano plot was generated to visualise the results, displaying the $\log_2$ fold change on the x-axis and the $-\log_{10}$ FDR-adjusted *p*-value on the y-axis.

*Linear discriminant analysis (LDA) effect size (LEfSe)* To complement DESeq2-based DA testing, we additionally applied linear discriminant analysis (LDA) using the LEfSe algorithm to identify microbial features that consistently discriminate between *S. mansoni*−infected and uninfected individuals[62]. While DESeq2 provides robust statistical inference and effect size estimates for individual taxa across groups, LDA ranks taxa based on their ability to explain group differences by combining

statistical significance with biological consistency and effect relevance. This approach helps prioritise taxa most likely to contribute to distinguishing phenotypic states.

Taxa with a logarithmic LDA score > 2.0 and *p* < 0.05 were considered significantly enriched. Analyses were stratified by community type (rural vs. urban) to account for environmental and lifestyle heterogeneity. Visualisation of LEfSe results was done via bar plots showing LDA scores.

By integrating both DA and LDA approaches, we capture a broader perspective on microbiome differences−identifying statistically robust changes (via DESeq2) while also highlighting microbial signatures with high discriminatory power (via LEfSe) that may serve as candidate biomarkers.

## Microbiome−CVD risk associations

To evaluate the direct associations between microbial taxa and specific CVD risk factors (e.g., blood pressure, total cholesterol, LDL cholesterol), multivariate linear regression models were fitted separately for infected and uninfected groups within rural and urban settings. Models were adjusted for age, sex, BMI, and diet (in the rural population). Full model outputs (effect sizes, SEs, 95% CIs, p-values) are provided (supplementary table 2 and 3). Significance was set at *p* < 0.05, and false discovery rate (FDR) correction was applied. The top 50 most abundant taxa were prioritised for analysis. Results were visualised with heatmaps and annotated by significance level (*p* < 0.05, *p* < 0.01).

## Mediation analysis

To investigate potential microbial mediation of the relationship between *S. mansoni* infection and CVD risk, non-parametric bootstrap-based mediation analysis was conducted using the mediation analysis function in pingouin python library[63]. This approach estimated the indirect effect of differentially abundant taxa (from differential abundance analysis (FDR-adjusted *p* < 0.05)) on CVD risk scores, with 5000 bootstrap iterations and 97.5% confidence intervals. Microbes demonstrating significant negative or positive mediation effects (*p* < 0.05) were visualised using alluvial plots to capture the pathway from infection status to cardiovascular outcome through the microbiome.

We applied a prespecified causal-inference framework to estimate the extent to which gut microbiome features mediated the association between *Schistosoma mansoni* infection and cardiometabolic risk traits. We constructed a directed acyclic graph (DAG; Supplementary Fig. 8) to formalise prior biological and epidemiological knowledge and to determine the minimally sufficient adjustment set required to block back-door confounding paths. Running the adjustmentSets() function on this DAG identified age, sex, and setting as the minimal confounder set for estimating both total effects and microbiome-mediated effects. Accordingly, all mediation analyses adjusted exclusively for these variables.

For each cardiometabolic outcome, we fitted counterfactual-based causal mediation models that partitioned the total effect of *S. mansoni* infection into natural direct effects and natural indirect effects operating through individual microbial taxa, analysed one mediator at a time.

## Metabolite−CVD risk associations

To investigate links between circulating metabolites and cardiometabolic traits, we fitted linear regression models for each metabolite using the combined dataset of rural and urban participants. Each model included the metabolite as the exposure and the cardiovascular risk factor of interest (systolic blood pressure, diastolic blood pressure, total cholesterol, or LDL cholesterol) as the outcome. The models were adjusted for age, sex, BMI, and *S. mansoni* infection status. Because rural and urban environments differ markedly in lifestyle and

ecological exposures, we additionally included a setting-by-metabolite interaction term to assess whether metabolite–CVD associations differed across environments.

For each metabolite, we extracted the effect estimate, standard error, 95% confidence intervals, and p-value for the association with the cardiovascular risk factor. Metabolites demonstrating statistical evidence of association at p < 0.01 were taken forward to multi-omics integration alongside CVD-associated microbes.

## Metabolomic profiling and pathway enrichment analysis

Untargeted metabolomics was conducted using liquid chromatography–mass spectrometry (LC-MS) on faecal samples. Differential metabolite abundance between *S. m+* and *S. m-* individuals was assessed using volcano plots with thresholds set at $p \leq 0.05$ and log 2-fold change (FC) $\geq 1.0$ (for upregulation in infected) or FC < 1.0 (for upregulation in uninfected). To evaluate whether the metabolomic profiles could discriminate between groups, we employed Partial Least Squares Discriminant Analysis (PLS-DA) using the caret package in R. The dataset was split into training and test sets using stratified sampling to maintain class balance. Model training was performed using 5-fold cross-validation, repeated three times, to optimise model parameters and assess classification performance. Model accuracy, sensitivity, and specificity were calculated on the test set, and the discriminative capacity was further evaluated by constructing Receiver Operating Characteristic (ROC) curve**s** and calculating the area under the curve (AUC) with 95% confidence intervals using the pROC package[64]. We assessed group-level differences in metabolomic profiles using PERMANOVA on scaled Euclidean distances. Feature importance was derived to identify the most discriminative metabolites.

Pathway enrichment analysis was conducted using Integrated Molecular Pathway Level Analysis (IMPaLA)[65], incorporating Kyoto Encyclopaedia of Genes and Genomes (KEGG), Reactome, and other curated databases. Samples were grouped according to specified criteria provided. For statistical analysis, a minimum fold change between sample groups of at least 1.5-fold, ANOVA *p* values of <0.05 were used in IMPaLA.

## Metabolite annotation and functional classification

Detected features were matched against reference libraries using mass-to-charge ratio (m/z) and retention time (RT) as primary identifiers. To improve matching precision, m/z values were rounded to five decimal places and RTs to one decimal place, generating a unique combined feature ID for each metabolite. These IDs were used to merge the detected features with annotation outputs from the xMSannotator platform, which provides multi-parameter chemical identification including adduct patterns, isotope distributions, and pathway associations. Annotation confidence was further refined by cross-referencing putative matches with the Human Metabolome Database. When available, we prioritised annotations with the highest confidence scores as assigned by the xMSannotator workflow. Significantly altered metabolites associated with insulin resistance and other cardiometabolic risk factors were annotated into functional classes using LIPID MAPS and CLASSYFIRE The annotated metabolites were grouped by chemical class and displayed via bar plots (for CLASSYFIRE categories) and pathway clusters (via RaMP-DB)[66]. The significance of pathway enrichment was determined using Fisher's exact test with multiple testing correction. Results were visualised via bar plots and redundancy-aware "lollipop" plots generated using RaMP-DB, clustering functionally overlapping pathways shown in Supplementary Figs. 9-13.

## Multimodal integration

Where relevant, co-association networks and integrative heatmaps were generated to explore relationships between *S. mansoni* infection, microbial taxa, metabolites, and CVD phenotypes. Correlation networks were built using the Spearman's rank correlation in the mixOmics package in R[67].

## Reproducibility

All microbiome and metabolomics analysis pipelines, including quality control thresholds, normalisation procedures, and statistical parameters, are described in sufficient detail to enable reproduction of the analyses. All analyses were performed in R (version 4.1.3)[68]. Complete R session information, including software versions, package dependencies, and computational environment details, is provided in Supplementary File.

## Reporting summary

Further information on research design is available in the Nature Portfolio Reporting Summary linked to this article.

## Data availability

The raw microbiome sequencing data generated in this study have been deposited in the NCBI BioProject database under accession code PRJNA1405921. The raw mass spectrometry–based faecal metabolomics data, together with relevant experimental metadata, have been deposited in the Metabolomics Workbench repository under Study ID ST004547 (Data track ID: 6961 and are assigned the digital object identifier https://doi.org/10.21228/M8Z255. The processed microbiome and metabolomics data generated in this study have been deposited in the Zenodo repository under the (https://doi.org/10.5281/zenodo.18186512) (https://doi.org/10.5281/zenodo.18186512). This DOI represents all versions of the dataset and will always resolve to the most recent version. The repository includes raw 16S rRNA gene sequencing data, processed microbiome data (including taxonomy assignments and abundance tables), and processed faecal metabolomics data. The raw sequencing data are provided in accordance with participant consent and applicable data protection regulations. The de-identified individual participant data that underlie the results reported in this article, including demographic information and other covariates, together with a data dictionary are stored in the LSHTM Data Compass repository under (https://doi.org/10.17037/DATA.00004919). Researchers who wish to access these data may submit a request through LSHTM Data Compass, detailing the data requested, the intended use, and evidence of relevant experience. Requests will be reviewed by the corresponding author(s) in consultation with the MRC/UVRI and LSHTM Uganda Research Unit Data Management Committee, with oversight from the UVRI and LSHTM ethics committees. Approved datasets will be provided with pseudonymised participant identifiers, enabling linkage to the microbiome and metabolomics datasets while maintaining participant confidentiality. Access is subject to execution of an appropriate data sharing agreement. A reporting summary for this article is available as a Supplementary Information file.

## Code availability

All scripts used for data processing, statistical analysis, and figure generation are available via Zenodo under the DOI 10.5281/zenodo.18186513 (https://doi.org/10.5281/zenodo.18186513).The analyses were performed using a combination of R and Python scripts together with established bioinformatics software for microbiome, metabolomics, and multi-omics integration analyses. A full list of all software packages used, including version numbers and their analytical purpose, is provided in the Supplementary Information.

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

## Acknowledgements

We are grateful to Dr Gyaviira Nkurunungi, Ms Joy Kabagenyi and Mr Alfred Ssekagiri for their expert comments on the manuscript. BW is partially supported by GCRF collaborative Grant (R120442) from the Royal Society awarded to Professors Richard Grencis and Alison Elliott, and is also partially funded by the National Institute for Health Research (NIHR) under its Global Health Research Group on Vaccines for Vulnerable People in Africa (VAnguard) (Grant Reference Number: NIHR134531), using UK aid from the UK Government to support global health research. This project has also been supported by Wellcome Trust Investigator Award Z10661/Z/18/Z and the Wellcome Centre for Cell Matrix Research Grant 088785/Z/09/Z awarded to Professor Richard Grencis, and Wellcome Trust (grant number 095778) awarded to Professor Alison Elliott. The views expressed in this publication are those of the author(s) and not necessarily those of the NIHR or the UK Government. The MRC/UVRI and LSHTM Uganda Research Unit is jointly funded by the UK Medical Research Council (MRC) and the UK Department for International Development (DFID) under the MRC/DFID Concordat agreement. The funders were not involved in the conceptualisation of the study, writing of the paper and the decision to submit it for publication.

## Author contributions

B.W. Conceived the study, designed the experiments, conducted the parasitological, microbiome and metabolomics conducted the primary statistics and bioinformatics analyses, interpreted the results, prepared the figures, and wrote the first draft of the manuscript. M.A.E.L. Contributed to the microbiome and metabolomics experiments and analyses and contributed to critical manuscript revision. A.J.B. supported the microbiome and metabolomics experiments, and manuscript draughting and revision. J.N. Contributed to the parasitological, microbiome and metabolomics experiments, and contributed to manuscript revision. D.K.T. Contributed to metabolomics data annotation and offered guidance on how metabolomics analysis was done. G.T. Provided specialist support in metabolomics experiments and pathway analysis contributed to critical manuscript review. R.E.S. Oversaw clinical coordination of parent projects, supported data acquisition, and provided critical input on the interpretation of the cardiovascular risk outcomes. E.L.W. Supervised statistical analysis for the study, advised on analytical strategy, and contributed to manuscript review and editing. D.P.K. Provided supervision of project, contributed to and reviewed the manuscript. R.K.G. provided senior supervision of the project, funding acquisition for this study, contributed to study conceptualisation and provided critical manuscript revisions. A.M.E. Provided senior supervision throughout the study, supported study design of the parent projects, funding acquisition of both the parent and current studies, advised on epidemiological methods, and contributed to critical revision of the manuscript. All authors reviewed and approved the final manuscript.

## Competing interests

The authors declare no competing interests.

## Additional information

[1]Immunomodulation and Vaccines Group, Vaccine Research Theme, Medical Research Council /Uganda Virus Research Institute and London School of Hygiene and Tropical Medicine (MRC/UVRI & LSHTM) Uganda Research Unit, Entebbe, Uganda. [2]Department of Immunology & Molecular Biology, School of Biomedical Science, College of Health Sciences, Makerere University, Kampala, Uganda. [3]Department of Infection Biology, London School of Hygiene and Tropical Medicine, London, United Kingdom. [4]The Lydia Becker Institute of Immunology and Inflammation and the Manchester Cell-Matrix Centre, Faculty of Biology, Medicine and Health, University of Manchester, Manchester, United Kingdom. [5]Manchester Institute of Biotechnology, Department of Chemistry, University of Manchester, Manchester, United Kingdom. [6]Biological Mass Spectrometry Core Facility, Faculty of Biology, Medicine and Health, University of Manchester, Manchester, UK. [7]Emerging and Re-emerging Infectious Diseases Unit, African Population and Health Research Centre, Nairobi, Kenya. [8]MRC International Statistics and Epidemiology Group, Department of Infectious Disease Epidemiology, London School of Hygiene and Tropical Medicine, London, United Kingdom. [9]Department of Clinical Research, London School of Hygiene and Tropical Medicine, London, United Kingdom. [10]These authors contributed equally: David P. Kateete, Richard K. Grencis, Alison M. Elliott. ✉e-mail: bridgious.walusimbi@mrcuganda.org

