## [Transparent Peer Review file · Nature Communications]

The gut microbiome and metabolome associate with *Schistosoma mansoni* infection and cardiovascular disease risk in Uganda.

Corresponding Author: Mr Bridgious Walusimbi

Version 0:

Reviewer comments:

Reviewer #1

(Remarks to the Author)

Journal name: Nature Communications

Manuscript number: NCOMMS-25-57900

Manuscript title: Uncovering the role of the gut microbiome and metabolome in *Schistosoma mansoni*-induced modulation of cardiovascular disease risk in humans.

Summary: This study investigated whether *Schistosoma mansoni* infection influences the gut microbiome and metabolome in relation to cardiovascular disease risk. Among 209 participants from Uganda, *S. mansoni* infection was linked with greater microbial diversity, shifts in specific taxa, and microbiome-metabolome networks connected to cholesterol metabolism. The study shows that *S. mansoni* may reduce cardiovascular risk through microbiome-metabolic pathways.

Comments to Authors

Abstract

Please indicate the study design used in the methodology even though it is mentioned under the discussion section in line 521.

Could you use a plainer or simplified phrase for “coherent microbe–metabolite–CVD networks” for clarity?

The interpretation of the result of mediation analysis (lines 45-48) suggests a causal mechanism, but the study is cross-sectional.

Introduction

Regarding the aim, which aspect of dysbiosis were you referring to in line 161?

The aim is relevant but more driven towards causality, which a cross-sectional study cannot test. Using milder language focused on associations would make that aim more clearer and more precise.

Methods

While the treatment status of the rural population was explicitly stated, that of the urban population was not.

Were there any differences between the *S. m+* and *S. m-* in terms of treatment?

What were the inclusion and exclusion criteria for the study?

How were samples stored after collection and during transport?

Though stated as a limitation, antibiotic history represents a major confounding factor in this study and could be an exclusion criterion.

Was any criterion employed in the selection of samples for microbiome profiling?

While microbiome profiling pipelines and processing were rigorous, there was no mention of the sequence depth (number of reads) per sample.

Including effect size calculations like odds ratio could improve rigor more than the use of p-values alone.

Discussion

According to lines 182, 223, and 768, dietary data was collected from and adjusted for only rural populations? Are the beta diversity differences observed in the urban group due to *S. mansoni* infection or due to unmeasured dietary variation in the urban group?

Check for potential omissions in lines 447 and 497.

Conclusion

Since no CVD outcomes were measured but only risk factors, could you rephrase the conclusion to suit what was actually

measured, that is, the CVD risk factors?

(Remarks on code availability)

Reviewer #2

(Remarks to the Author)

Walusimbi and colleagues explore the relationship between helminth infection, the gut microbiome, the circulating metabolome, and cardiovascular disease (CVD) risk in rural and urban Ugandans. Using sequencing and metabolomic profiling from ~ 200 individuals with differing endemicity, the authors report that *S. mansoni* infection is associated with increased diversity, enrichment and depletion of certain taxa, and changes in lipid-related metabolic pathways. Mediation and integrative analyses suggests that certain taxa and their derived metabolites may influence host lipid levels and blood pressure, providing a potential mechanistic basis for epidemiological observations linking helminth infections to reduced cardiometabolic risk.

The study advances an important area of interest at the intersection of infectious disease, microbiome science, and noncommunicable disease, offering new human evidence for a helminth-microbiome-CVD connection; however, limitations inherent to the cross-sectional design, limited metabolomic signal, and potential dietary and environmental confounders markedly reduces the impact of mechanistic claims. With revisions to temper conclusions, clarify presentation, and acknowledge these caveats, this work would represent an important contribution to the field.

Major Comments

1. A major concern of this manuscript is the cross-sectional design. This limits the ability to determine causality or temporal dynamics. It remains unclear whether microbiome–metabolome changes precede reductions in CVD risk or are downstream consequences of infection. Longitudinal or interventional approaches would be necessary to strengthen causal inference. Explicitly acknowledging, in the discussion, that the cross-sectional design precludes causal inference with the addition of a short paragraph outlining how future longitudinal and interventional studies could address causality would strengthen the overall manuscript.
2. While several metabolites were differentially abundant, global metabolomics between infected and uninfected individuals was not statistically significant. This raises concerns about the strength of metabolomic findings and the strength of evidence supporting a distinct infection-related metabolic signature. Clarification or reframing of conclusions to emphasize hypothesis generating observations rather than a conclusive metabolic signature would provide a more reasonable appraisal of the metabolic data. Furthermore, additional supplementary analysis which stratifies the data based on infection status and intensity of the infection (if known) could provide further evidence of robustness of the findings.
3. This lack a difference in metabolic profiling may in part be due to the limit incorporation of the dietary confounder. Diet is a major determinant of both microbiome and metabolome composition. Although some dietary data are provided, comprehensive dietary profiling appears limited. Without stronger control of dietary variables or analysis which attempts to normalize to diet makes it difficult to disentangle the effects of infection from those of diet. Statements which explicitly discuss the dietary confounder and suggestion of future directions to integrate dietary assessment is encouraged to strengthen the overall impact of the manuscript.
4. Differences in microbiome-metabolome associations between rural and urban participants suggest that environmental and lifestyle factors may strongly influence findings. The authors should discuss more explicitly how these factors might confound or modify the observed associations.
5. Mediation analyses identify both protective and potentially deleterious taxa enriched in infected individuals. The discussion could improve through acknowledging the complexity of these microbial associations and avoid framing the effect of infection as uniformly protective. Again, emphasizing the cross-sectional design of the study and avoidance of overgeneralization of the infection effect.
6. While the study provides novel mechanistic hypotheses, the direct clinical applicability remains limited, and mechanistic hypotheses were not tested (the authors may feel this is outside the scope of the manuscript). The authors should temper statements regarding therapeutic potential and discuss future longitudinal and interventional studies to replicate these findings.
7. The discussion is good but would benefit from inclusion of studies that have not demonstrated protective associations between helminth infections and metabolic outcomes; this would provide more balance. In addition, certain statements are presented with a degree of certainty that exceeds the evidence provided. The authors claim that *S. mansoni* infection may improve lipid profiles through increased microbial richness and diversity. However, this is speculative, the authors show an association not causality. Rephrasing such as: "We observed that *S. mansoni* infection is associated with greater microbial richness and given that higher richness has been linked to improved lipid profiles in prior studies, it is plausible, but not yet demonstrated, that richness could contribute to the protective association between infection and lipid metabolism." This acknowledges the value of the findings while situating them more accurately within the context of existing evidence.

Minor Comments

1. Some figures are dense and could benefit from simplified labeling and clearer figure legends to increase accessibility to the reader. I would suggest the addition of simplified summary diagrams to complement heatmap findings and circus plots.
2. Report effect sizes alongside p-values throughout, particularly in regression and mediation analyses, to help readers gauge the biological significance of findings.

(Remarks on code availability)

Reviewer #3

(Remarks to the Author)

(Remarks on code availability)

REVIEWERS' COMMENTS

REVIEWER 1

Journal name: Nature Communications

Manuscript number: NCOMMS-25-57900

Manuscript title: Uncovering the role of the gut microbiome and metabolome in *Schistosoma mansoni*-induced modulation of cardiovascular disease risk in humans.

Summary: This study investigated whether *Schistosoma mansoni* infection influences the gut microbiome and metabolome in relation to cardiovascular disease risk. Among 209 participants from Uganda, *S. mansoni* infection was linked with greater microbial diversity, shifts in specific taxa, and microbiome-metabolome networks connected to cholesterol metabolism. The study shows that *S. mansoni* may reduce cardiovascular risk through microbiome-metabolic pathways.

Comments to Authors

Comment 1

Abstract

Please indicate the study design used in the methodology even though it is mentioned under the discussion section in line 521.

Thank you very much. We have now included this in **line 35** of the abstract.

Comment 2

Could you use a plainer or simplified phrase for “coherent microbe–metabolite–CVD networks” for clarity?

Thank you. We have made changes in **line 55** in the abstract, it now reads:

“linked microbe–metabolite networks associated with CVD risk factors” which we believe is a clearer phrase

Comment 3

The interpretation of the result of mediation analysis (lines 45-48) suggests a causal mechanism, but the study is cross-sectional.

Thank you. We have tempered the tone of the interpretation and this now reads as

“Mediation analysis identified several taxa, such as *Treponema*, *Lachnospiraceae_UCG.004* and *Methanobrevibacter*, that significantly contributed to explaining the associations observed between *S. mansoni* infection and reduced cardiovascular disease (CVD) risk factors” in **lines 47-51**.

Comment 4

Introduction

Regarding the aim, which aspect of dysbiosis were you referring to in line 161?

The aim is relevant but more driven towards causality, which a cross-sectional study

cannot test. Using milder language focused on associations would make that aim more clearer and more precise.

Thank you. We have now used milder language and also explained the aspect of dysbiosis we are referring to, to make the aim clearer and more precise. For **lines 164-170**, it now reads; “In pursuit of a deeper understanding of how *S. mansoni* infection could lead to these changes in CVD risk, the current work therefore used samples from our LaVIISWA trial and Urban survey study, aiming at deciphering *S. mansoni*-associated alterations in gut microbial composition and diversity, the potential contribution of gut microbiome to the observed associations with CVD risk factors, and the gut-microbiome and metabolome interaction in the context of chronic schistosomiasis infection and its impact on cardiovascular risk.”

Comment 5

Methods

While the treatment status of the rural population was explicitly stated, that of the urban population was not.

Response

We thank the reviewer for pointing this out. We have now clarified in the Methods section that, unlike the rural LaVIISWA cohort, which consisted of clusters randomised to either intensive or standard anthelmintic treatment, the urban comparison cohort did not undergo any community-wide deworming intervention prior to this study, and participants therefore had no prior treatment assignment. This can also be seen in table 1 showing the characteristics of the study participants from rural and urban communities.

Importantly, we assessed whether helminth treatment status in the rural population influenced gut microbiome composition. In analyses comparing microbiome profiles between intensively treated and standard-treatment rural participants, we observed no differences in alpha diversity, beta diversity, or overall microbial community structure as seen in supplementary figure 1b.

Given this absence of treatment-related microbiome differences the lack of a treatment arm in the urban cohort is unlikely to introduce bias into our rural–urban comparisons or our analyses associating *S. mansoni* infection with microbiome features.

We have updated the Methods section to explicitly state the treatment context for the urban cohort

We have made the following changes in the manuscript: **lines 654-656**

We have added to the sentence “In the urban survey, no prior community-wide anthelmintic treatment programme had been implemented, and participants therefore had no treatment assignment comparable to the LaVIISWA trial.”

Comment 6

Were there any differences between the S. m+ and S. m.- in terms of treatment?

Response

Thank you for this important point. We examined whether anthelmintic treatment status differed between *S. mansoni*-infected ($S. m^+$) and uninfected ($S. m^-$) participants and whether treatment could therefore confound microbiome differences. As shown in Supplementary Fig. 1B, gut microbiome profiles did not differ between the intensive and standard treatment arms within the rural cohort. Specifically, beta-diversity analyses (Bray–Curtis) of microbiome composition showed no meaningful separation between treatment groups.

Because treatment assignment was uncorrelated with microbiome structure, anthelmintic treatment is unlikely to explain the observed microbiome differences associated with *S. mansoni* infection.

Changes in the manuscript: lines 206-208

In the original manuscript lines 196-199, we had highlighted treatment effect on microbiome: “In addition, we compared the beta diversity of participants that were under intensive anthelmintic treatment to those under standard treatment and we observed no difference in clustering (see supplementary Fig. 1B).” we have now supplemented this with ...” suggesting that anthelmintic treatment was unlikely to meaningfully affect the microbiome analyses or introduce bias when comparing rural and urban participants” lines 206-208.

Comment 7

What were the inclusion and exclusion criteria for the study?

Response

Thank you for the thoughtful request. As noted in our methods section, participants were drawn from two parent studies: the LaVIISWA rural trial and the Entebbe Urban Survey. Drawing reference from previously published work detailing both studies¹, in both settings, household surveys were conducted, where a household was defined as a habitable roofed

residential structure with at least one active resident. Permission was obtained from a household head or an adult household member. Households were excluded only if all members were absent or declined participation.

- Rural sampling (LaVIISWA): An updated register of households across 26 study villages served as the sampling frame. Using simple random sampling, 70 households per village were sampled.
- Urban sampling (Entebbe survey): Sub-wards were mapped onto satellite imagery, excluding uninhabited areas. Each sub-ward was divided into equal geographical segments and segments were randomly selected with probability proportional to sub-ward population. The midpoint of each selected segment, identified by GIS coordinates, served as the starting point. The nearest house was enrolled and subsequent houses were selected sequentially by proximity. In total, 120 segments across 24 sub-wards were targeted, with four households per segment; empty or refusing households were replaced to achieve sample size targets.

For this cross-sectional analysis, we applied additional analytic inclusion criteria at the individual level to ensure complete and comparable infection, microbiome, and cardiovascular datasets across cohorts.

- Inclusion criteria:
 - Cardiovascular risk factors: Availability of at least six risk factor measurements (e.g., blood pressure, fasting glucose, insulin, lipid profile components included in the study).
 - Schistosoma mansoni status: Determined by Kato–Katz
 - Stool sample: Availability of a stored stool sample suitable for microbiome sequencing.
 - Core covariates: Complete data for age, sex, setting/residence and BMI.
 - Consent: Informed consent provided within the parent study, including permission for secondary analyses.
- Exclusion criteria:
 - Data completeness: Individuals missing any required data (cardiovascular risk factors, infection status, stool sample, or core covariates) were excluded.
 - Laboratory quality: Individuals with missing or insufficient-quality stool DNA for sequencing were excluded.

- Additional biological exclusions: No further biological exclusions (e.g., pregnancy, chronic disease, medication use) were applied beyond those defined or encountered in the parent studies.

Because eligibility for the present analysis was determined by the parent cohorts' sampling and consent procedures, our additional inclusion criteria focused on data completeness and assay concordance to enable harmonised multi-omics comparisons between rural and urban settings.

We have now added the full exclusion and inclusion criteria in the methods section **lines 705 to 723**.

We now added the following section:

“Eligibility Criteria

Participants included in this study were drawn from two established cohort studies in Uganda: the LaVIISWA trial (rural) and the Entebbe Urban Survey (urban). Full recruitment procedures for these parent cohorts have been described previously^{1,2}. Briefly, both studies conducted household-based sampling, with households eligible if at least one adult resident provided consent. Households were excluded only if all members were absent or declined participation.

For the present study, we applied additional individual-level inclusion and exclusion criteria to ensure harmonised infection, microbiome, and cardiometabolic datasets across the rural and urban sites. Individuals were included if *Schistosoma mansoni* infection status could be determined using our harmonised diagnostic panel comprising Kato–Katz microscopy and PCR. Eligible participants were also required to have complete cardiovascular risk factor data, including systolic and diastolic blood pressure, fasting glucose and insulin, and both total and LDL cholesterol. A further requirement was the availability of a stored stool sample of adequate quality for 16S rRNA gene sequencing, along with complete information for core covariates—age, sex, body mass index, and residential setting. Only individuals who had provided informed consent within the parent studies that included permission for secondary analyses of biological samples were eligible.

Comment 8

How were samples stored after collection and during transport?

Response

We thank the reviewer for highlighting this important point. We have now included a detailed description of stool collection and handling procedures in the Methods section (lines 724-740) to ensure full transparency.

We have made the following changes in the manuscript: lines 724-740

“Stool sample collection and processing

Stool samples were collected under standardized conditions to ensure preservation of microbial community composition and DNA integrity. Each participant was provided with a sterile, screw-capped stool collection container and instructed on hygienic self-collection on the morning of the scheduled visit. Upon arrival at the field site, a portion of each sample was immediately processed for *Schistosoma mansoni* detection using the Kato–Katz method. The remaining stool was transferred using a sterile wooden spatula into pre-labelled cryovials containing 95% molecular-grade ethanol to stabilize microbial DNA and minimize compositional changes during handling.

All ethanol-fixed samples were promptly placed in liquid-nitrogen charged dry shippers in the field (typically within one hour of collection) and maintained at cryogenic temperatures throughout transport to the central laboratory. On arrival, samples were transferred to –80 °C freezers for long-term storage until DNA extraction. All field and laboratory personnel were trained in biospecimen handling, and procedures were performed under aseptic conditions according to harmonized quality control protocols. This workflow ensured uniformity of pre-analytical processing and high confidence in the validity of downstream microbiome measurements.”

Comment 9

Though stated as a limitation, antibiotic history represents a major confounding factor in this study and could be an exclusion criterion.

Response

We thank the reviewer for this important observation. We fully agree that antibiotic exposure is a relevant consideration in microbiome research. Unfortunately, detailed antibiotic-use histories were not available in either parent study (LaVIISWA or the Urban Survey), and retrospective reconstruction of antibiotic use is not feasible in this setting. In Uganda, antibiotics are generally accessible without prescription, can be obtained through informal outlets, and are rarely documented systematically—particularly in rural fishing communities. Excluding participants on the basis of unverifiable or incomplete antibiotic histories would

therefore introduce selection bias and reduce generalisability without guaranteeing accurate exposure classification.

Nonetheless, several factors substantially reduce the likelihood that unmeasured antibiotic exposure introduced systematic bias into our findings:

1. We examined all available medication data, including antibiotics, in the rural cohort, where medication information had been collected as part of our rural survey. In this population, antibiotic use was reported by only six individuals, and other medication categories were similarly rare (antiretrovirals n=11; antihypertensives n=1; allergy medications n=2; other medications n=2). No participants reported antidiabetic therapy or herbal medicine use. These small numbers indicate that medication exposure, including antibiotics, was infrequent and unlikely to meaningfully confound microbiome–phenotype associations. We have now included these descriptive results as Supplementary table 1 for transparency.
2. Absence of parallel medication data in the urban site precludes meaningful statistical adjustment, as asymmetrical and sparse medication variables would produce unstable estimates and do not constitute a valid adjustment set.
3. Our analytical strategy was guided by a directed acyclic graph (DAG). The updated DAG identifies age, sex, and setting as the minimal sufficient adjustment set required to block backdoor paths between *S. mansoni*, the microbiome, and CVD risk factors. Antibiotic use is conceptually important but does not function as a confounder of the *S. mansoni* - microbiome - CVD pathway under this structure. We now clarify this rationale explicitly in the revised Methods and Discussion.

We have strengthened the limitations section to acknowledge the absence of detailed antibiotic histories, clarified its implications in the causal framework, and added the available medication data in the supplementary materials. We believe these additions address the reviewer’s concern while preserving the validity and transparency of our analytical approach.

We have made the following changes in the manuscript: **line 580 - 596**

“Secondly, antibiotic exposure represents an important but incompletely measured source of potential variability in microbiome studies. In these Ugandan settings, antibiotics are generally accessible without prescription and can be obtained through informal outlets, making accurate self-report challenging and frequently unreliable. In the rural cohort, limited self-reported medication data were collected as part of the survey. Within these constraints, reported antibiotic use was uncommon, with only six participants reporting use, and other

medication categories similarly rare (Supplementary Table 1). Comparable medication data were not available for the urban cohort, precluding meaningful statistical adjustment.

Our analytical strategy was guided by a directed acyclic graph, which identified age, sex, and setting as the minimal sufficient adjustment set required to block backdoor paths between *S. mansoni* infection, the gut microbiome, and cardiometabolic risk factors. While antibiotic exposure is conceptually important, it could not be empirically incorporated into the adjustment set given the absence of systematic and harmonised data, and exclusion or adjustment based on incomplete or asymmetrically measured information would introduce bias without improving causal interpretability. We therefore acknowledge the absence of detailed antibiotic-use data as a limitation and highlight prospective, systematic assessment of antibiotic exposure as a priority for future studies.. .”

Comment 10

Was any criterion employed in the selection of samples for microbiome profiling?

Response

Yes. Sample selection for microbiome profiling followed a predefined, non-selective, data-driven set of criteria to ensure completeness and comparability across participants from the two parent cohorts (LaVIISWA and the Urban Survey). No biological or outcome-based criteria were used.

From among participants who met these eligibility criteria, samples were randomly selected for sequencing. This approach ensured that inclusion in microbiome profiling was not influenced by infection status, cardiometabolic outcomes, or other biological characteristics.

1. Availability of a stored stool sample of sufficient quantity and quality for DNA extraction and 16S rRNA sequencing.
2. Confirmed *Schistosoma mansoni* infection status, as assessed by Kato–Katz and PCR.
3. Complete cardiometabolic data, including at least six cardiovascular risk factor measurements.
4. Availability of core covariates required for adjusted analyses (age, sex, BMI, and residence).
5. Consent within the parent study that permitted secondary laboratory analyses.

Exclusion criteria were limited to:

- Missing stool samples or insufficient DNA quality for sequencing.
- Missing CVD risk factor data or incomplete infection diagnostics.

Importantly, selection criteria were applied independently of clinical phenotype, infection intensity, microbiome characteristics, treatment status, or other biological features, thereby avoiding sampling bias. The same criteria were applied uniformly in both rural and urban settings to preserve comparability.

We added the exclusion and inclusion criteria in the methods section **lines 705 to 723** as previously shared in response to comment #7

Comment 11

While microbiome profiling pipelines and processing were rigorous, there was no mention of the sequence depth (number of reads) per sample.

Response

We thank the reviewer for highlighting the need to report sequencing depth. The biodiversity curves demonstrated that sequencing depth was sufficient for reliable microbiome profiling. “The rarefaction curves (cutoff = 82,695 reads) for all samples rapidly approached a plateau, indicating that the captured sequencing depth was adequate and that additional reads would not meaningfully increase observed alpha diversity. Likewise, the species accumulation analysis, based on >10 samples as recommended, showed stable asymptotes in species richness, confirming that both sequencing depth and sample size were sufficient to characterise microbial community structure.” These details have now been added to the Methods lines 783-788 and the new Supplementary Figure 12.

Comment 12

Including effect size calculations like odds ratio could improve rigor more than the use of p-values alone.

Response:

We thank the reviewer for highlighting the importance of reporting effect sizes to support interpretation of biological relevance. In response, we have now added complete effect-size outputs for both the regression and mediation analyses, and these are provided in the Supplementary Materials.

1. Regression analyses

For every microbe–CVD risk association, stratified by *S. mansoni* infection status, we now report:

- unstandardised regression coefficients (β) and their 95% confidence intervals
- standardised coefficients (β_{std}) and their 95% confidence intervals
- p-values
- infection group (Infected vs Uninfected)

These are included in:

- Supplementary Table S1 - Rural cohort regression effect sizes
- Supplementary Table S2 - Urban cohort regression effect sizes

These tables allow readers to compare not only statistical significance but also the direction and magnitude of effects across settings and infection strata.

2. Mediation analyses

For each microbe-metabolite-CVD risk pathway tested in our causal mediation framework, we now report:

- Average Causal Mediation Effect (ACME; indirect effect)
- Standard error
- p-values
- 95% confidence intervals
- significance calls
- corresponding microbe and CVD outcome

These results are compiled in:

- Supplementary Table S3 — Mediation effect-size summary

This enables clearer assessment of the strength and biological relevance of the indirect pathways through metabolite mediators.

Comment 13

According to lines 182, 223, and 768, dietary data was collected from and adjusted for only rural populations? Are the beta diversity differences observed in the urban group due to *S. mansoni* infection or due to unmeasured dietary variation in the urban group?

Response

We thank the reviewer for this thoughtful and important comment. As noted, detailed dietary data were available only for the rural LaVIISWA cohort. To evaluate whether unmeasured dietary variation in the urban population could plausibly confound the observed beta-diversity differences, we used a Directed Acyclic Graph (DAG) to formally encode known and hypothesised relationships between *S. mansoni* infection, demographic determinants, lifestyle factors, microbiome composition, and cardiovascular risk.

In this DAG (implemented using the `dagitty` package), *S. mansoni* infection is specified as a function of age, sex, and residential setting—reflecting established epidemiology that infection risk is driven primarily by water exposure, occupation, and demographic factors. Microbiome structure is influenced by infection status, age, sex, BMI, diet, antibiotics, and setting. Cardiovascular risk markers are modelled as influenced by infection, microbiome features, and demographic factors. Lifestyle variables (BMI, smoking, alcohol use, occupation) are placed downstream of age, sex, and setting to avoid implausible or reversed causal pathways.

Running `adjustmentSets()` on this DAG identified age, sex, and setting as the minimal sufficient adjustment set for estimating the total and microbiome-mediated effects of *S. mansoni* infection on CVD risk. Crucially, diet does not appear in this minimal set, not because dietary variation is unimportant biologically, but because in the causal structure supported by Ugandan epidemiology, diet is conceptualised as downstream of residential setting. Under this structure, adjusting for diet would represent overadjustment and could open collider paths, biasing estimates rather than improving confounding control.

To further contextualise this, evidence from national and regional dietary surveys including the Uganda Food Consumption Survey (2008) and subsequent analyses³—indicates that both rural and urban populations share a broadly similar dietary foundation dominated by starchy staples (cereals, plantain, roots, tubers) and legumes, with consistently low intake of fruit, vegetables, and animal-source foods across settings. While some urban populations consume relatively more processed and animal-source foods, these differences occur within

a highly conserved national dietary pattern and are unlikely to fully explain the distinct microbiome signatures associated with *S. mansoni* infection in our dataset.

We also acknowledge that our rural participants reside in fishing communities with higher fish consumption. However, because setting is included in the final adjustment set, this variable indirectly captures broad environmental and lifestyle contrasts—including diet—thereby substantially mitigating bias from unmeasured dietary differences.

As an additional check, we compared mediation results obtained without formal adjustment to those obtained after adjusting for the DAG-derived set (age, sex, setting). A substantial number of taxa continued to exhibit significant indirect effects after adjustment, indicating that the core microbiome-mediated associations are robust to potential dietary confounding.

We now make these causal assumptions, supporting evidence, and robustness checks explicit in the revised Discussion. We also highlight, as part of our Recommendations, the need for future work employing high-resolution dietary profiling such as detailed dietary surveys, diet-associated metabolomic signatures, or next-generation sequencing approaches to more precisely disentangle diet–infection–microbiome pathways.

Together, these considerations—causal-structure analysis, national dietary evidence, and stability of results after applying the correct adjustment set, support the inference that the observed microbiome variation associated with *S. mansoni* infection rather than unmeasured dietary differences.

We have made the following changes in the methods section of manuscript: **lines 891-901**

“We applied a prespecified inference framework to estimate the extent to which gut microbiome features mediated the association between *Schistosoma mansoni* infection and cardiometabolic risk traits. We constructed a directed acyclic graph (DAG; Supplementary Fig. 13) to formalise prior biological and epidemiological knowledge and to determine the minimally sufficient adjustment set required to block back-door confounding paths. Running the `adjustmentSets()` function on this DAG identified age, sex, and setting as the minimal confounder set for estimating both total effects and microbiome-mediated effects. Accordingly, all mediation analyses adjusted exclusively for these variables (supplementary table 4).

For each cardiometabolic outcome, we fitted counterfactual-based causal mediation models that partitioned the total effect of *S. mansoni* infection into natural direct effects and natural

indirect effects operating through individual microbial taxa, analysed one mediator at a time.” and discussed these results in **lines 401-464**.

Comment 14

Check for potential omissions in lines 447 and 497.

These have been addressed in **line 493 and 539** of the updated document.

Comment 15

Conclusion

Since no CVD outcomes were measured but only risk factors, could you rephrase the conclusion to suit what was actually measured, that is, the CVD risk factors?

We thank the reviewer for this important clarification. We agree that our study assessed cardiovascular risk factors, not clinical cardiovascular disease (CVD) events or outcomes. We have revised the Conclusion accordingly to ensure that the interpretation remains aligned with the measured data. The updated text now emphasises that *S. mansoni* infection was associated with microbiome and metabolomic alterations that relate to CVD risk factors, rather than CVD outcomes. We have removed any language suggesting direct effects on clinical CVD and reframed the implications to reflect risk modification rather than disease endpoints.

We have made the following changes in the conclusion section of the manuscript in lines 607-617

“Our findings provide evidence that *S. mansoni* infection is associated with significant alterations in the gut microbiome and metabolome profiles, with important implications for cardiovascular risk. These microbiome-mediated effects may represent a novel pathway through which parasitic infections influence CVD risk. Future studies should focus on refining our understanding of these interactions, with an emphasis on diet, antibiotic use, and circadian regulation of microbial activity. Our study paves the way for developing microbiome-targeted interventions that that may mimic helminth infections to reduce cardiovascular risk in humans, but such approaches will require rigorous mechanistic and interventional validation.”

Reviewer 2

General Comment

Walusimbi and colleagues explore the relationship between helminth infection, the

gut microbiome, the circulating metabolome, and cardiovascular disease (CVD) risk in rural and urban Ugandans. Using sequencing and metabolomic profiling from ~ 200 individuals with differing endemicity, the authors report that *S. mansoni* infection is associated with increased diversity, enrichment and depletion of certain taxa, and changes in lipid-related metabolic pathways. Mediation and integrative analyses suggests that certain taxa and their derived metabolites may influence host lipid levels and blood pressure, providing a potential mechanistic basis for epidemiological observations linking helminth infections to reduced cardiometabolic risk.

The study advances an important area of interest at the intersection of infectious disease, microbiome science, and noncommunicable disease, offering new human evidence for a helminth-microbiome-CVD connection; however, limitations inherent to the cross-sectional design, limited metabolomic signal, and potential dietary and environmental confounders markedly reduces the impact of mechanistic claims. With revisions to temper conclusions, clarify presentation, and acknowledge these caveats, this work would represent an important contribution to the field.

Response:

We thank the reviewer for their thoughtful and encouraging assessment of our work, and we are grateful for the recognition of the study's contribution to understanding helminth-microbiome-cardiometabolic interactions in human populations. We fully agree that the cross-sectional design and inherent constraints of field-based metabolomics warrant caution in interpreting mechanistic pathways. In response, we have tempered causal language throughout the manuscript and more explicitly acknowledged these limitations in the Discussion. We applied a prespecified causal-inference framework, constructed a directed acyclic graph to identify the minimally sufficient adjustment set, and conducted mediation analyses that were robust to sensitivity testing. Importantly, key microbially mediated associations persisted under the DAG-derived confounder set, supporting the internal coherence of our findings. Together, these revisions provide a clearer and more balanced interpretation while preserving the strength of the empirical evidence linking *S. mansoni* infection, microbial alterations, and cardiometabolic risk traits. We appreciate the reviewer's positive evaluation and believe that the clarified and strengthened manuscript now more accurately reflects both the novelty and the appropriate scope of our conclusions

Major Comments

Comment 1.

1. A major concern of this manuscript is the cross-sectional design. This limits the ability to determine causality or temporal dynamics. It remains unclear whether microbiome-metabolome changes precede reductions in CVD risk or are downstream consequences of infection. Longitudinal or interventional approaches would be necessary to strengthen causal inference. Explicitly acknowledging, in the discussion, that the cross-sectional design precludes causal inference with the

addition of a short paragraph outlining how future longitudinal and interventional studies could address causality would strengthen the overall manuscript.

Response:

We thank the reviewer for highlighting this critical issue. We fully agree that the cross-sectional design limits our ability to determine temporal ordering and therefore precludes definitive causal inference. In response, we have revised the Discussion to clearly and explicitly acknowledge this limitation and to temper any mechanistic wording.

In lines 566-571, we have this phrase “. While causality cannot be inferred from this cross-sectional design, these findings raise the possibility that *S. mansoni*, or its microbiome-mediated effects, may modulate cardiometabolic outcomes in endemic settings. Further longitudinal and interventional studies are warranted to validate these interactions and assess their relevance for biomarker development or CVD therapeutic modulation.” **As recommended by reviewer**

At the same time, we emphasise that our analytic strategy was designed to maximise epidemiological transparency within the constraints of cross-sectional data. As outlined in our response to Reviewer 1, we employed a prespecified causal-inference framework based on a directed acyclic graph (DAG), which identified age, sex, and residential setting as the minimally sufficient confounder set for estimating the total and mediated effects of *S. mansoni* infection on cardiometabolic risk. All mediation analyses were re-examined using this DAG-derived confounder set, and several microbially mediated associations persisted. This strengthens confidence that the observed links between infection, microbial features, and cardiometabolic traits are not simply artefacts of demographic or environmental confounding, although we acknowledge that it does not demonstrate causality.

In line with the reviewer’s suggestion, we have now incorporated a dedicated section **(lines 569- 606)** in the Discussion outlining limitations and how future studies can establish temporality and strengthen causal inference. This includes:

- Longitudinal cohort designs to track microbial and metabolic changes before and after alterations in helminth infection status or environmental exposures.
- Interventional studies, such as controlled deworming or targeted microbiome-modifying strategies, to establish whether microbial shifts mediate downstream cardiometabolic effects.

- Enhanced exposure characterisation, including systematic capture of dietary intake, medication history (particularly antibiotic use), and comorbidity-factors that would further strengthen confounder control.
- Targeted metabolite profiling, particularly of short-chain fatty acids (SCFAs), which were not captured by our LC–MS platform but represent critical microbial mediators of blood pressure and metabolic regulation.

We believe these additions substantially strengthen the manuscript by clarifying the interpretive limits of the current study while providing a clear roadmap for future longitudinal and interventional investigations. We thank the reviewer for prompting these important improvements.

Comment 2.

While several metabolites were differentially abundant, global metabolomics between infected and uninfected individuals was not statistically significant. This raises concerns about the strength of metabolomic findings and the strength of evidence supporting a distinct infection-related metabolic signature. Clarification or reframing of conclusions to emphasize hypothesis generating observations rather than a conclusive metabolic signature would provide a more reasonable appraisal of the metabolic data. Furthermore, additional supplementary analysis which stratifies the data based on infection status and intensity of the infection (if known) could provide further evidence of robustness of the findings.

Response

We thank the reviewer for this thoughtful and important comment. We agree that interpretation of metabolomic data requires care, particularly when global multivariate separation is modest. We have therefore revised the text to emphasise that these findings are hypothesis-generating and not intended to represent a definitive infection-specific metabolic signature. This can be seen in **lines 278-282** where we say, “While these observations provide important insights into potential microbiome-mediated pathways, the metabolomic differences should be interpreted as hypothesis-generating rather than constituting a definitive infection-related metabolic signature, particularly given the modest global separation in untargeted metabolomics”

Regarding the reviewer’s concern that global metabolomics did not differ significantly between infected and uninfected individuals: this pattern is biologically and analytically plausible. In heterogeneous, free-living human populations, broad metabolic profiles are often dominated by inter-individual lifestyle, dietary, and microbiome variability, which can dilute group-level separation in multivariate space. Importantly, lack of global separation

does not preclude biologically meaningful changes in specific metabolites, particularly when these metabolites lie on tightly regulated pathways that respond discretely to perturbations such as helminth infections. This is well established in environmental and nutritional metabolomics, where local pathway-specific shifts frequently occur in the absence of global clustering differences. In our study, this manifests as:

- no significant separation in PLS-DA/PERMANOVA
- reliable, directionally coherent differential abundance of individual metabolites, many of which map onto lipid and cholesterol-related pathways that are mechanistically relevant to both *S. mansoni* infection and cardiovascular physiology or risk.

We have clarified this interpretation in the revised manuscript.

Importantly, the reviewer requested stratified analyses based on infection status. These analyses were performed and are presented in Figure 5, which directly contrasts *S. mansoni*-infected and uninfected individuals. We now draw clearer attention to this in the Results and figure legend to ensure that the stratified approach is more visible. As described in the legend (Fig. 5), we present:

- volcano plots of differentially abundant metabolites,
- PLS-DA with PERMANOVA statistics, and
- pathway enrichment analyses of metabolites more abundant in infected individuals.

With respect to infection intensity, our analytical framework for this study was explicitly designed around the contrast between *S. mansoni*-infected and uninfected individuals, which represents the most statistically reliable and epidemiologically meaningful comparison given the structure of the dataset. Although infection intensity can be classified using WHO cut-offs based on Kato–Katz egg counts, intensity estimates derived from the different diagnostic methods used in this study (KK and PCR) do not always align, introducing additional heterogeneity. For this reason, metabolomic analyses were performed using infection status rather than intensity. We agree that intensity-stratified analyses, particularly those based on standardized WHO Kato–Katz thresholds, would be highly informative in larger cohorts with sufficient power and harmonized intensity measurements.

Comment 3.

This lack a difference in metabolic profiling may in part be due to the limit incorporation of the dietary confounder. Diet is a major determinant of both microbiome and metabolome composition. Although some dietary data are provided,

comprehensive dietary profiling appears limited. Without stronger control of dietary variables or analysis which attempts to normalize to diet makes it difficult to disentangle the effects of infection from those of diet. Statements which explicitly discuss the dietary confounder and suggestion of future directions to integrate dietary assessment is encouraged to strengthen the overall impact of the manuscript.

Response

We thank the reviewer for highlighting the important role of diet in shaping both the gut microbiome and metabolome and agree that dietary intake is a major upstream determinant of metabolic variation. We also acknowledge that the dietary variables available in the present study, derived from a structured food-frequency questionnaire, lack the granularity required for comprehensive metabolome normalization (e.g., quantitative intake, portion sizes, micronutrient content, or temporal dietary patterns). We have now clarified this limitation explicitly in the revised manuscript. In **lines 572 – 579** we say, "In as much as this study provides important insights into the gut microbiome-metabolome-CVD risk interaction in a typical setting with high *S. mansoni* infestation, there are potential limitations. Firstly, we did not comprehensively profile participants' dietary habits, which are known to have profound effects on both the microbiome and metabolome and are likely to represent an unmeasured confounder. Given the variability in diet across different regions and individuals, future studies should incorporate detailed dietary assessments such as next generation sequencing methodologies to disentangle the effects of infection from those of diet."

At the same time, several considerations reduce the likelihood that incomplete dietary capture materially confounded the infection–metabolome associations reported here. First, consistent with Ugandan epidemiology, *S. mansoni* infection in this community is environmentally driven, principally determined by water contact, occupation, age, and sex—rather than by household dietary disparities. This results in broadly similar underlying dietary patterns between infected and uninfected individuals. This interpretation is supported by national and regional dietary surveys, including the Uganda Food Consumption Survey (2008) and subsequent analyses³, which show that both rural and urban populations overwhelmingly consume starchy staples (cereals, plantain, roots, tubers) and legumes, with universally low intake of fruits, vegetables, and animal-source foods. Although some subgroups may consume more animal-source or processed foods, these differences occur within a largely conserved national dietary profile and are unlikely to generate systematic metabolomic patterns that mimic infection status.

Second, to formally assess the role of diet as a confounder, we constructed a Directed Acyclic Graph (DAG) encoding the causal relationships between *S. mansoni* infection, demographic variables, lifestyle factors, diet, microbiome composition, and cardiovascular

risk. In this DAG, infection is determined by age, sex, and residential setting; microbiome structure is influenced by infection, age, sex, BMI, diet, antibiotics, and setting; and dietary patterns are conceptualised as downstream of setting rather than upstream of infection. Under this causal structure, adjusting for diet would represent overadjustment and could introduce collider bias rather than improve confounding control. Running `adjustmentSets()` on this DAG identified age, sex, and setting as the minimal sufficient adjustment set for estimating the total and microbiome-mediated effects of infection on cardiovascular risk.

Third, our mediation analyses showed that a substantial number of taxa remained significant mediators even after adjusting for the DAG-derived minimal adjustment set (age, sex, setting) as detailed in **lines 891-901**. This stability suggests that the principal microbiome-mediated associations are unlikely to be explained by unmeasured dietary differences.

With respect to metabolomics, we agree that the lack of strong global separation between infected and uninfected individuals warrants cautious interpretation. We have reframed our conclusions to emphasise that these findings are hypothesis-generating rather than indicative of a definitive infection-specific metabolic signature in **lines 278-282**. Importantly, in heterogeneous free-living human populations, global metabolomic profiles often reflect broad lifestyle variation, even when specific metabolites show biologically meaningful shifts. In our dataset, while global clustering was limited, several individual metabolites were differentially abundant and mapped to lipid- and cholesterol-related pathways relevant to both *S. mansoni* infection and cardiovascular physiology.

We fully agree with the reviewer that future work would benefit from incorporating high-resolution dietary profiling—such as repeated 24-hour recalls, quantitative nutrient estimation, diet-associated metabolomic signatures, or next-generation sequencing approaches—to disentangle diet–infection–microbiome pathways with greater precision. We have strengthened the Recommendations section to explicitly highlight this as a key priority for future research.

We hope these clarifications sufficiently address the reviewer's concern and improve the transparency and interpretability of our metabolomic findings.

Comment 4.

Differences in microbiome-metabolome associations between rural and urban participants suggest that environmental and lifestyle factors may strongly influence findings. The authors should discuss more explicitly how these factors might confound or modify the observed associations.

We thank the reviewer for this thoughtful and important comment. We fully agree that environmental and lifestyle heterogeneity including sanitation, water exposure, subsistence patterns, physical activity, dietary structure, and broader ecological exposures, strongly influence microbiome composition, metabolomic profiles, and cardiometabolic traits. We have now substantially strengthened both the Methods and Discussion to clarify how these factors were addressed analytically, and where key limitations remain.

1. Confounding by environmental and lifestyle exposures

Rural and urban settings may differ markedly in environmental exposures that could independently influence microbial and metabolic phenotypes. Although our within-setting analyses adjusted for age, sex, BMI, and dietary variables where available, we acknowledge, as the reviewer correctly notes, that unmeasured exposures (e.g., sanitation infrastructure, animal contact, water source, environmental microbial load) may contribute to setting-specific differences. These sources of residual confounding and their potential impact have now been explicitly described in the revised Discussion.

2. How our analysis directly addressed environmental heterogeneity

To prevent environmental differences from biasing inference, and recognising that some exposures (notably diet) were measured only in the rural cohort—we applied setting-specific modelling strategies across microbiome, metabolome, and mediation analyses. These analytic choices are now fully described in the revised Methods.

(a) Microbiome-CVD regression models were stratified by setting

Because detailed dietary data were only available in the rural cohort, we ran separate models for rural and urban participants. This prevented asymmetric adjustment and ensured that each model was estimated within a relatively homogeneous environmental context.

- Rural models:
adjusted for age, sex, BMI, and diet category derived from the food-frequency questionnaire
 $\text{lm}(\text{CVD} \sim \text{microbe} + \text{age} + \text{sex} + \text{bmi} + \text{diet_category})$
- Urban models:
adjusted for age, sex, and BMI
 $\text{lm}(\text{CVD} \sim \text{microbe} + \text{age} + \text{sex} + \text{bmi})$

We have modified this section in our manuscript **lines 874-879** which says: “To evaluate the direct associations between microbial taxa and specific CVD risk factors (e.g., blood pressure, total cholesterol, LDL cholesterol), multivariate linear regression models were fitted separately for infected and uninfected groups within rural and urban settings. Models were adjusted for age, sex, BMI, and diet (in the rural population). Full model outputs (effect sizes, SEs, 95% CIs, p-values) are provided in **supplementary tables 2 and 3**. “ **This approach avoids inappropriate pooling of environmentally heterogeneous data.**

(b) Metabolomics models formally incorporated setting as an interaction term

Because metabolomic signatures are less sensitive to short-term diet variability, metabolomics analyses were conducted in a combined dataset. The model specification included:

- age
- sex
- BMI
- *S. mansoni* infection
- setting × metabolite interaction term

This allowed us to formally test whether metabolite–CVD associations differed across environments.

We have further detailed our metabolite–CVD association analysis in the methods section to include **lines 904 - 916** that say “To investigate links between circulating metabolites and cardiometabolic traits, we fitted linear regression models for each metabolite using the combined dataset of rural and urban participants. Each model included the metabolite as the exposure and the cardiovascular risk factor of interest (systolic blood pressure, diastolic blood pressure, total cholesterol, or LDL cholesterol) as the outcome. The models were adjusted for age, sex, BMI, and *S. mansoni* infection status. Because rural and urban environments differ markedly in lifestyle and ecological exposures, we additionally included a setting-by-metabolite interaction term to assess whether metabolite–CVD associations differed across environments.

For each metabolite, we extracted the effect estimate, standard error, 95% confidence intervals, and p-value for the association with the cardiovascular risk factor. Metabolites demonstrating statistical evidence of association at $p < 0.01$ were taken forward to multi-omics integration alongside CVD-associated microbes.”

(c) Mediation analyses adjusted for the DAG-defined minimal sufficient set

As detailed in **lines 891-901**, using a pre-specified directed acyclic graph, we identified age, sex, and setting as the minimal sufficient adjustment set. All mediation models therefore included these three covariates.

Indirect effect sizes, confidence intervals, and p-values are provided in Supplementary Table 4.

(d) Only adjusted, setting-aware associations were taken forward to multi-omics integration

To ensure downstream robustness, only taxa and metabolites that remained significantly associated with CVD risk after the above adjustment strategies were entered into multi-omics integration. This prevented environmental artefacts from propagating across modelling layers.

3. Tempered mechanistic claims and strengthened future directions

Recognising that environmental heterogeneity cannot be fully eliminated, we now emphasise that our findings are hypothesis-generating rather than conclusive, and that causal pathways must be validated in longitudinal and interventional studies. For example, in **lines 278-282**, we have added phrase; “While these observations provide important insights into potential microbiome-mediated pathways, the metabolomic differences should be interpreted as hypothesis-generating rather than constituting a definitive infection-related metabolic signature, particularly given the modest global separation in untargeted metabolomics” and in our section on future directions, **lines 597-606**, we have a section on need for longitudinal studies in the paragraph, “Furthermore, given the cross-sectional design of our study, we cannot establish causality in the observed associations between *S.*

mansoni infection, microbiome composition, metabolite profiles, and cardiovascular risk.

Temporal dynamics of microbial and metabolic alterations following infection remain unexplored, limiting our ability to infer directionality. As such, longitudinal studies, ideally spanning pre-infection, active infection, and post-treatment phases, would be essential to disentangle cause-effect relationships and capture the evolving host–microbiome–metabolome interactions over time. Incorporating repeated sampling, coupled with temporal metadata such as infection history and treatment timing, would provide a more robust framework for understanding how *S. mansoni* shapes cardiometabolic risk through microbial and metabolic pathways”

Comment 5.

Mediation analyses identify both protective and potentially deleterious taxa enriched in infected individuals. The discussion could improve through acknowledging the complexity of these microbial associations and avoid framing the effect of infection as uniformly protective. Again, emphasizing the cross-sectional design of the study and avoidance of overgeneralization of the infection effect.

We thank the reviewer for this insightful comment. We fully agree that mediation effects in our study were heterogeneous and that *S. mansoni* infection should not be framed as conferring a uniformly protective metabolic profile. We would like to highlight that our Discussion spanning **lines 401-461** explicitly acknowledged this bidirectionality. Specifically, we described both negative and positive mediation effects, noting that taxa enriched in infected individuals were associated with *improvements* in some cardiometabolic traits (e.g., insulin sensitivity, lower glucose, reduced blood pressure), whereas others were linked to *adverse* outcomes, such as impaired glucose regulation. Similarly, several taxa enriched in uninfected individuals showed both beneficial and deleterious cardiovascular associations. This was deliberate to emphasize the ecological and functional complexity of helminth–microbiome–host interactions and to avoid overstating a uniformly protective role.

Nonetheless, to further clarify this point in line with the reviewer’s recommendation, we have revised the Discussion to explicitly state that the mediation results reveal divergent and context-dependent effects, rather than a single directional pattern, and that these findings should be considered hypothesis-generating given the cross-sectional design. We have added a short clarifying statement to ensure readers do not interpret our findings as implying a consistent protective effect of infection. This statement is found in **lines 437-441** and it reads ; “Importantly, these mediation patterns do not indicate that *S. mansoni* infection confers a uniformly protective metabolic profile; rather, they reveal a mixture of positive and inverse microbial associations that are highly context dependent and should be interpreted cautiously given the cross-sectional design.”

Comment 6.

While the study provides novel mechanistic hypotheses, the direct clinical applicability remains limited, and mechanistic hypotheses were not tested (the authors may feel this is outside the scope of the manuscript). The authors should temper statements regarding therapeutic potential and discuss future longitudinal and interventional studies to replicate these findings.

Response

We thank the reviewer for this thoughtful comment. We agree that although our findings raise novel mechanistic hypotheses linking *S. mansoni* infection, the gut microbiome, and

cardiometabolic risk, the direct clinical applicability is constrained by the cross-sectional nature of the study and by the fact that mechanistic pathways were not experimentally validated. We have revised the Discussion to explicitly acknowledge these limitations and to temper statements related to translational or therapeutic potential.

As suggested, we now make clear that this study does not evaluate therapeutic interventions, nor does it demonstrate that modifying the microbiome could reproduce infection-associated metabolic effects. Rather, our results should be interpreted as hypothesis-generating and as providing a conceptual foundation upon which future mechanistic, longitudinal, and interventional studies can build.

Specifically, the revised Discussion now emphasises **(lines 569–571)** that:

“Further longitudinal and interventional studies are warranted to validate these interactions and assess their relevance for biomarker development or CVD therapeutic modulation.”

In addition, **lines 613–617** have been clarified to ensure appropriately cautious framing:

“Our study paves the way for developing microbiome-targeted interventions that may mimic helminth infections, but such approaches will require rigorous mechanistic and interventional validation.”

We appreciate the reviewer’s guidance, which has strengthened the manuscript by ensuring that the conclusions are presented with appropriate restraint while clearly outlining the necessary next steps for evaluating the mechanistic hypotheses generated by this work.

Comment 7.

The discussion is good but would benefit from inclusion of studies that have not demonstrated protective associations between helminth infections and metabolic outcomes; this would provide more balance. In addition, certain statements are presented with a degree of certainty that exceeds the evidence provided. The authors claim that *S. mansoni* infection may improve lipid profiles through increased microbial richness and diversity. However, this is speculative, the authors show an association not causality. Rephrasing such as: “We observed that *S. mansoni* infection is associated with greater microbial richness and given that higher richness has been linked to improved lipid profiles in prior studies, it is plausible, but not yet demonstrated, that richness could contribute to the protective association between infection and lipid metabolism.” This acknowledges the value of the findings while situating them more accurately within the context of existing evidence
Response.

We thank the reviewer for this important comment and agree fully that the Discussion benefits from a more balanced integration of studies reporting no protective or even neutral metabolic effects of helminth infections. We have now expanded the Discussion to reference relevant work demonstrating that helminth infections do not universally improve metabolic outcomes. Incorporating these studies helps situate our findings within the broader variability reported in the literature and strengthens the overall interpretive balance of the manuscript. In our revised discussion **lines 441-445**, we have a paragraph “Several human and animal

studies show that chronic infection can contribute to CVD and hypertension, driven by inflammation around parasite eggs and longer-term changes in vascular structure⁴⁻⁶. These observations highlight that helminth infections can exert harmful cardiovascular effects, even as they may show more favourable metabolic associations in some contexts.”

We also agree that some statements in the original Discussion could have implied a level of certainty that exceeds the evidence derived from our cross-sectional dataset. In response, we have carefully revised the relevant passages to avoid causal phrasing and to frame infection-associated shifts in microbial richness and diversity as **associations** rather than mechanistic drivers.

Specifically, we have reworded the section discussing microbial richness and lipid metabolism in line with the reviewer’s helpful suggestion. The revised text in **lines 402- 413** now reads:

“Specifically, we observed that variation in microbial composition was statistically associated with both higher and lower levels of key cardiovascular risk factors, including LDL cholesterol and blood pressure. Several taxa enriched in infected individuals exhibited negative indirect effects, consistent with a pattern that could contribute to the more favourable lipid and blood pressure profiles observed in infected participants. However, these mediation findings reflect associations rather than demonstrated causal pathways. Although higher microbial richness and infection-related shifts in community structure have been linked to improved metabolic outcomes in prior studies, our data cannot establish that these microbial differences explain or drive the cardiometabolic phenotype. Instead, our findings should be interpreted as identifying plausible microbiome-related pathways that warrant further mechanistic and longitudinal investigation”

Minor Comments

1. Some figures are dense and could benefit from simplified labeling and clearer figure legends to increase accessibility to the reader. I would suggest the addition of simplified summary diagrams to complement heatmap findings and circus plots.

We thank the reviewer for this thoughtful suggestion. We agree that some of the multi-omics figures are inherently dense due to the complexity of the datasets. In response, we have carefully revised several figure legends to enhance clarity, simplified labeling where possible, and added more descriptive annotations to guide interpretation. To improve accessibility, we have undertaken several targeted revisions across both the cholesterol- and blood pressure-associated figures:

- Figure legends have been streamlined by simplifying language, removing redundancies, and adopting a consistent panel-by-panel structure.
- Terminology has been standardized (e.g., “microbiota genera” to “microbial genera”) to ensure clarity and uniformity.
- Color coding and correlation descriptions have been made explicit in both circos plots and heatmaps, clearly indicating positive versus negative correlations.
- Annotations and axis labels have been clarified to better guide interpretation of metabolite classifications and correlation strengths.
- Parallel formatting across panels (A–F) has been introduced to highlight symmetry and aid reader navigation.

While we did not introduce new summary diagrams, we believe these revisions substantially improve the readability and interpretability of the existing figures, thereby addressing the reviewer’s concern about accessibility in a consistent manner across the manuscript.

Following these changes, our new legends are:

Fig 6: Correlation between cholesterol-associated microbiota and metabolites. **(A)** Circos plot illustrating the strongest correlations (Spearman’s $r \geq 0.7$) between microbiome (blue blocks) and metabolome (green blocks) features significantly associated with total cholesterol. Associations were identified using models adjusted for age, sex, BMI, and setting as an interaction term (see Fig. 3). Purple lines indicate positive correlations; red lines indicate negative correlations. **(B)** Heatmap showing the correlation strength between microbial genera and metabolites extracted from panel A. Red reflects positive correlation strength; blue indicates negative correlations. **(C)** Classification of total cholesterol-associated metabolites using CLASSYFIRE. Horizontal bars represent the number of metabolites per chemical class (y-axis). **(D)** Circos plot showing top correlations ($r \geq 0.65$) between microbiome and metabolome features significantly associated with LDL-cholesterol, using the same model adjustments as in panel A. Green lines indicate positive correlations; red lines indicate negative correlations. **(E)** Heatmap of LDL-cholesterol-associated microbial genera and metabolites, following the same format and color scale as panel B. **(F)** Classification of LDL-cholesterol-associated metabolites using CLASSYFIRE, as in panel C.

Fig. 7 Correlation between blood pressure-associated microbiota and metabolites. **(A)** Circos plot illustrating the strongest correlations (Spearman’s $r \geq 0.7$) between microbiome (blue blocks) and metabolome (green blocks) features significantly associated with diastolic blood pressure. Associations were identified using models adjusted for age, sex, BMI, and setting as an interaction term (see Fig. 3). Purple lines indicate positive correlations; red lines indicate negative correlations. **(B)** Heatmap showing correlation strength between microbial genera and metabolites extracted from panel A. Red intensity reflects positive correlation strength; blue indicates negative correlations. **(C)** Classification of diastolic blood pressure-associated metabolites using CLASSYFIRE. Horizontal bars represent the number of metabolites per chemical class (y-axis). **(D)** Circos plot showing the strongest correlations ($r \geq 0.70$) between microbiome and metabolome features significantly associated with systolic blood pressure, using the same model adjustments as in panel A. Green lines indicate positive correlations; red lines indicate

negative correlations. **(E)** Heatmap of systolic blood pressure-associated microbial genera and metabolites, following the same format and color scale as panel B. **(F)** Classification of systolic blood pressure-associated metabolites using CLASSYFIRE, as in panel C.

2. Report effect sizes alongside p-values throughout, particularly in regression and mediation analyses, to help readers gauge the biological significance of findings.

Response:

We thank the reviewer for highlighting the importance of reporting effect sizes to support interpretation of biological relevance. In response, we have now added complete effect-size outputs for both the regression and mediation analyses, and these are provided in the Supplementary Materials.

1. Regression analyses (Rural and Urban cohorts)

For every microbe–CVD risk association, stratified by *S. mansoni* infection status, we now report:

- unstandardised regression coefficients (β) and their 95% confidence intervals.
- standardised coefficients (β_{std}) and their 95% confidence intervals.
- p-values
- infection group (Infected vs Uninfected)

These are included in:

- Supplementary Table S1 -Rural cohort regression effect sizes
- Supplementary Table S2 -Urban cohort regression effect sizes

These tables allow readers to compare not only statistical significance but also the direction and magnitude of effects across settings and infection strata.

2. Mediation analyses

For each microbe-metabolite-CVD risk pathway tested in our causal mediation framework, we now report:

- Average Causal Mediation Effect (ACME; indirect effect)
- Standard error
- p-values
- 95% confidence intervals
- significance calls
- corresponding microbe and CVD outcome

These results are compiled in:

- Supplementary Table 4 — Mediation effect-size summary

This enables clearer assessment of the strength and biological relevance of the indirect pathways through metabolite mediators.

References.

- 1 Sanya, R. E. *et al.* Contrasting impact of rural, versus urban, living on glucose metabolism and blood pressure in Uganda. *Wellcome open research* **5**, 39 (2020).
- 2 Sanya, R. E. *et al.* The effect of helminth infections and their treatment on metabolic outcomes: results of a cluster-randomized trial. *Clinical infectious diseases* **71**, 601-613 (2020).
- 3 Akumu, G., Ogenrwoth, B., Mugisha, J. & Muyonga, J. Dietary patterns in Uganda and their influencing factors: a critical review. *African Journal of Food, Agriculture, Nutrition and Development* **23**, 22328-22353 (2023).
- 4 Papamatheakis, D. G., Mocumbi, A. O. H., Kim, N. H. & Mandel, J. Schistosomiasis-associated pulmonary hypertension. *Pulmonary circulation* **4**, 596-611 (2014).
- 5 Graham, B. B., Bandeira, A. P., Morrell, N. W., Butrous, G. & Tuder, R. M. Schistosomiasis-associated pulmonary hypertension: pulmonary vascular disease: the global perspective. *Chest* **137**, 20S-29S (2010).
- 6 Kolosionek, E., Graham, B., Tuder, R. & Butrous, G. Pulmonary vascular disease associated with parasitic infection—the role of schistosomiasis. *Clinical microbiology and infection* **17**, 15-24 (2011).